# Antibacterial Properties of Peptide and Protein Fractions from *Cornu aspersum* Mucus

**DOI:** 10.3390/molecules29122886

**Published:** 2024-06-18

**Authors:** Lyudmila Velkova, Aleksandar Dolashki, Ventsislava Petrova, Emiliya Pisareva, Dimitar Kaynarov, Momchil Kermedchiev, Maria Todorova, Pavlina Dolashka

**Affiliations:** 1Institute of Organic Chemistry with Centre of Phytochemistry, Bulgarian Academy of Sciences, Acad. G. Bonchev Str., bl. 9, 1113 Sofia, Bulgaria; aleksandar.dolashki@orgchm.bas.bg (A.D.); mitkokaynarov@abv.bg (D.K.); kermedchiew@yahoo.com (M.K.); krasimirova_m@yahoo.com (M.T.); 2Faculty of Biology, Sofia University, 8 Dragan Tzankov blvd., 1164 Sofia, Bulgaria; v.petrova@biofac.uni-sofia.bg (V.P.); episareva@uni-sofia.bg (E.P.); 3Businesslab Ltd., Acad. G. Bonchev Str., bl. 4A, 1113 Sofia, Bulgaria

**Keywords:** *Cornu aspersum* mucus, antimicrobial peptides, de novo MS/MS sequencing, proteomic analysis, mucus proteins with antimicrobial action

## Abstract

The discovery and investigation of new natural compounds with antimicrobial activity are new potential strategies to reduce the spread of antimicrobial resistance. The presented study reveals, for the first time, the promising antibacterial potential of two fractions from *Cornu aspersum* mucus with an MW < 20 kDa and an MW > 20 kDa against five bacterial pathogens—*Bacillus cereus* 1085, *Propionibacterium acnes* 1897, *Salmonella enterica* 8691, *Enterococcus faecalis* 3915, and *Enterococcus faecium* 8754. Using de novo sequencing, 16 novel peptides with potential antibacterial activity were identified in a fraction with an MW < 20 kDa. Some bioactive compounds in a mucus fraction with an MW > 20 kDa were determined via a proteomic analysis on 12% sodium dodecyl sulfate–polyacrylamide gel electrophoresis (SDS–PAGE) and bioinformatics. High homology with proteins and glycoproteins was found, with potential antibacterial activity in mucus proteins named aspernin, hemocyanins, H-lectins, and L-amino acid oxidase-like protein, as well as mucins (mucin-5AC, mucin-5B, mucin-2, and mucin-17). We hypothesize that the synergy between the bioactive components determined in the composition of the fraction > 20 kDa are responsible for the high antibacterial activity against the tested pathogens in concentrations between 32 and 128 µg/mL, which is comparable to vancomycin, but without cytotoxic effects on model eukaryotic cells of *Saccharomyces cerevisiae*. Additionally, a positive effect, by reducing the levels of intracellular oxidative damage and increasing antioxidant capacity, on *S. cerevisiae* cells was found for both mucus extract fractions of *C. aspersum*. These findings may serve as a basis for further studies to develop a new antibacterial agent preventing the development of antibiotic resistance.

## 1. Introduction

The World Health Organization (WHO) has reported an alarming increase in the number of bacterial strains resistant to conventional antibiotics, which is a threat to public health safety [1]. Additionally, antibiotic resistance (AR) is a global problem with important economic impacts [2,3]. The causes of AR are multifactorial, with the indiscriminate and prolonged use of antibiotics in both human and veterinary medicine as well as agriculture being paramount to the development and spread of drug-resistant microorganisms [4,5].

Bacteria are complex organisms and the effect of antibiotics on bacterial DNA is weak, but may provoke new acquired resistance mechanisms [6]. AR is a public health problem, spreading through humans, animals (domestic and wild), and the environment (water and air) [4]. Antibiotic resistance is therefore a silent pandemic that requires global health solutions [7]. The problem of antibiotic resistance necessitates an urgent search for alternatives to conventional antibiotics, with new modes of action and less susceptibility to bacterial resistance. 

Antimicrobial peptides (AMPs) may be one of the solutions with which to address this problem. These evolutionarily conserved peptides display a remarkable structural and functional diversity, and have been found in virtually all organisms, ranging from simple prokaryotes to complex eukaryotes, including humans [8,9]. AMPs are an important part of the innate immune systems of various organisms, providing a first line of defense against pathogenic invasion [10]. In contrast to antibiotics, AMPs have been shown to act on multiple targets on the plasma membranes of pathogenic bacteria as well as intracellular targets, and some of them have potent activity on drug-resistant bacteria [11]. AMPs from both synthetic and natural sources demonstrate broad-spectrum antimicrobial activity with high specificity and low toxicity; therefore, they are excellent candidates with which to overcome antibiotic resistance [11]. Studies reveal that cationic AMPs exert antibacterial activity by interacting with negatively charged bacterial membranes, resulting in increased membrane permeability, which leads to cell membrane damage, lysis, and cell content release, eventually resulting in cell death [11]. α-helical and β-sheet AMPs possess different binding mechanisms to bacterial membranes and can cause specific dynamic changes in membrane structures [12]. Furthermore, depending on peptide–lipid ratios, AMPs can be either vertically or parallel-oriented in membranes, causing membrane fragmentation or membrane pore formation, respectively [11]. Unlike cationic AMPs, the modes of action of anionic AMPs are not yet well understood, but it has been suggested that the antibacterial action of maximin H5 against *Staphylococcus aureus* is also related to membrane disruption [13]. Some AMPs at low concentrations can kill bacteria without changing membrane integrity [14]. Instead of directly interacting with the membrane, these AMPs kill bacteria by inhibiting important pathways inside a cell, such as DNA replication and protein synthesis, enzyme activity and cell wall synthesis, or by promoting the release of lyases to destroy cell structures [11,14]. 

AMPs play a crucial role against various infections in invertebrate species, which, unlike vertebrates, do not have an adaptive immune system, and, therefore, rely solely on innate defense mechanisms [15]. Invertebrate organisms of the phylum Mollusca represent one of the largest reservoirs of pharmacologically active compounds due to their great diversity (surpassed only by arthropods) and their ability to adapt to almost all types of habitats [16,17]. Bioactive compounds from hemolymph and mucus from gastropods have been shown to be powerful bioactive combinations with potential applications in combating pathogenic bacteria [18,19,20,21,22,23,24,25,26]. 

In the last decade, several studies reported antibacterial and antioxidant properties, as well as healing potential in treating wounds on the mucus of some land snails [18,21,22,23,26,27,28,29]. The antimicrobial activity of land snail mucus is known to be related not only to the presence of AMPs but also to some antibacterial proteins and glycoproteins [30,31]. A number of studies have reported that some proteins and glycoproteins in the mucus of various land snails (*Achatina fulica*, *H. aspersa*, *Cryptozona bistrialis*, *Lissachatina fulica,* and *Hemiplecta differenta*) are responsible for the antimicrobial properties of extracts from these snails [22,23,26,30,31,32,33,34].

The presented analyses in the current study are on new natural compounds with antimicrobial activity that upgrade our previous studies on the antibacterial activity of *Cornu aspersum* snail mucus. New information is provided on the antibacterial activity of two mucus fractions with an MW < 20 and an MW > 20 kDa from *C. aspersum* against five pathogenic bacteria (*Bacillus cereus* 1085, *Propionibacterium acnes* 1897, *Salmonella enterica* 8691, *Enterococcus faecalis* 3915, and *Enterococcus faecium* 8754) in addition to the identified important mucus components associated with antimicrobial activity.

## 2. Results

### 2.1. Preparation of Snail Mucus Extracts

The mucus was collected from *C. aspersum* snails growing on Bulgarian eco-farms using patented technology, without disturbing the biological functions of snails [19,21]. The resulting crude mucus extract was homogenized and centrifuged to remove coarse impurities. The purified native mucus extract was separated into two major fractions via ultrafiltration under pressure with polyethersulfone membrane filters with pore sizes of 20 kDa (Microdyn Nadir™ from the STERLITECH Corporation, Goleta, CA, USA). 

### 2.2. Analysis and Characteristics of the Isolated Mucus Fraction with an MW < 20 kDa

After purification via RP-HPLC on a BioSill C18 HL 90–10 column, the molecular masses of the bioactive compounds in the fraction with an MW < 20 kDa (Figure 1) were identified by mass spectrometry.

#### 2.2.1. Molecular Mass Analysis and De Novo Sequencing of Peptides by Mass Spectrometry

A thorough characterization of a fraction with an MW < 20 kDa is presented via matrix-assisted laser desorption/ionization time-of-flight mass spectrometry analyses (MADI-TOF-MS analyses) up to 3 kDa (Figure 2) and a range between 3 and 20 kDa (Figure 3). The MS spectrum presented in Figure 2 shows that the mucus fraction with an MW < 3 kDa contained various peptides with different masses in the region between 900 and 3011 Da. Peptides determined as protonated molecule ions [M+H]^+^ at *m*/*z* 1376 Da, 1438 Da, 1738 Da, 1796 Da, 1909 Da, 1966 Da, and 2292 Da dominate the MS spectrum (Figure 2).

Peptides and polypeptides with higher molecular weights detected in the fraction with an MW < 20 kDa are presented on the MS spectrum in Figure 3, as molecular protonated ions [M + H]^+^ at *m*/*z* 5573.742 Da, 6429.681 Da, 7337.199 Da, 11,143.605 Da, 12,464.216 Da, 15,059.328 Da, and 18,005.169 Da, as well as various other ions with lower intensities. The determined ions [M+H]^+^ at *m*/*z* 12.464 Da, 15,059.328 Da, and 18,005.169 Da are in good agreement with similar proteins in the snail mucus of *C. aspersum* [26,35], *Helix pomatia* [36], and *A. fulica* [37], reported previously.

The amino acid sequences (AASs) of low-molecular-weight peptides (with an MW < 3 kDa) were identified by de novo sequencing experiments (MS/MS analyses) of the [M+H]^+^ ions. Following b- and y-fragmentation ions in the MS/MS spectrum of peptide [M+H]^+^ at *m*/*z* 1966.11, the amino acid sequence LLLDNKGGGLVGGLLGGGGKGGG was identified (Figure 4).

Thus, the amino acid sequences of 16 new peptides with molecular masses below 3000 Da were identified, and the AASs of another 6 peptides were confirmed (Table 1). The determined physicochemical characteristics of the identified peptides, such as isoelectric points (pIs), the grand average of hydropathicity (GRAVY), and net charge by the ExPASy MW/pI tool program and ExPASy ProtParam tool are presented in Table 1. The analysis showed the presence of both cationic and anionic, as well as neutral, amphipathic peptide structures.

The analysis showed the presence of cationic, anionic, and neutral amphipathic peptide structures with generally hydrophobic surfaces, but nine hydrophilic peptides were also identified (nos. 1, 6–10, 12, and 15, Table 1). Based on the found primary structures of peptides (Table 1), their antimicrobial activity was predicted using iAMPpred software (http://cabgrid.res.in:8080/amppred) (accessed on 6 March 2024), an extensive database [41]. The results showed that peptides nos. 3, 7, 8, 13, 17, 18, 20, and 21–23 had the highest prognostic, antibacterial, and antifungal activities, while peptides 8 and 17 had the highest prognostic antiviral activities.

The alignment of the AASs of peptides shown in Table 1 with database AMPs via CAMPSing software (http://campsign.bicnirrh.res.in/, accessed on 24 March 2024) revealed similarity with known antimicrobial peptides (presented in Appendix A). Peptides nos. 10, 13, 17, 18, 20, 21, 22, and 23 show high homology with glycine-rich antimicrobial peptides, procambarin, holotricin, microcin B, acanthoscurrin and ctenidin; with the Gly/Leu-rich antimicrobial peptide leptoglycin; and glycine-rich protein GWK, while peptides nos. 7, 8, 12, and 15 revealed homology with defensin-like protein 196, glycine-rich protein GWK, crustin-like antimicrobial peptide, and different forms of gallinacin as well as shepherin (presented in Appendix A). 

The presented evidence confirmed that identified mucus peptides belong to the AMP family. Furthermore, the obtained results can be considered as basic information in the study of bioactive peptides from the *C. aspersum* mucus extract and for their potential biomedical applications.

#### 2.2.2. Characterization of the Mucus Extract Fraction with an MW > 20 kDa via Electrophoretic and MALDI-Tof-MS Analyses

The electrophoretic profile determined by 12% SDS-PAGE of the mucus extract fraction from *C. aspersum* with an MW > 20 kDa (at a concentration of 1.4 mg/mL) clearly shows various protein bands with MWs primarily in the region between 20 and 200 kDa (Figure 5a). The highest expression was observed for proteins with an MW of 17.6 kDa, 26.71 kDa, 29–32 kDa, 39.115 kDa, between 48 and 60 kDa, 97.0 kDa, and over 100 kDa (Figure 5a). Proteins with an MW of 20.5 kDa, 23.6 kDa, 29.1 kDa, 35.93 kDa, 70.89 kDa, and 84.77 kDa are also present in the composition of the fraction with an MW > 20 kDa, although with a lower expression.

A more accurate analysis of molecular masses and protein intensities in the mucus fraction with an MW > 20 kDa was obtained by ImageQuant^TM^ TL v8.2.0 software, after the scanning of 12% SDS-PAGE (Figure 5b,c). Based on the performed analysis, 18 protein bands were identified, most of which had an MW between 23 and 200 kDa. The obtained results presented in Figure 5c are in good agreement with the results of the MALDI-Tof-MS analysis in the 20–80 kDa region of fresh extract from *C. aspersum* mucus [29].

#### 2.2.3. Identification of Bioactive Compounds in the Mucus Extract Fraction with an MW > 20 kDa from *C. aspersum* Snails

Proteins in the fraction with an MW > 20 kDa from a mucus extract were determined after a search was performed in the UniProt database (https://www.uniprot.org, 3 March 2024) for proteins in molluscs and gastropods, as well as in published data for mucus proteins in snails with an MW corresponding to an electrophoretic analysis (Figure 5c). Based on the conducted search, it was found that the protein presented at protein band 17.565 kDa corresponded well with the mucus protein [*C. aspersum*, QEG59314] detected at 17.5 kDa in [26]. Detected proteins in the range 20–31 kDa probably belong to families of glutathione peroxidase (GPx) and glutathione transferases (GSTs) because several studies have confirmed the antioxidant properties (glutathione transferase activity) of *C. aspersum* mucus [27,28,29]. 

Results from the UniProt database show that the following proteins were detected in this region: glutathione peroxidases from *Biomphalaria glabrata* with an MW of 21.252 kDa (A0A2C9L5T6), an MW of 22.812 kDa (A0A2C9JW73), and an MW of 24.101 kDa (A0A2C9JBG3); peroxiredoxin-5 from *P. canaliculata* (A0A2T7NZ99, MW of 20.751 kDa); glutathione S-transferases with an MW of 27.978 kDa (from *P. canaliculata*, A0A2T7PWN7), with an MW of 27.599 kDa (from *B. glabrata*, A0A9U8DVA0), and with an MW of 27.774 kDa (from *Haliotis rubra*, XM_025244070.1); and probable glutathione S-transferase with an MW of 23.092 kDa (from *Aplysia californica*, XM_035968272.1). The proteins with an MW between 30 and 40 kDa are in good agreement with the identified proteins in [26], which suggests that some of them are lectins and are also related to the antimicrobial properties of mucus. Recently, Cerullo et al. reported a new protein class, termed conserved anterior mollusk proteins (CAMPs), detected in the mucus of *C. aspersum* with an MW < 40 kDa via a proteomic analysis on SDS-PAGE [34].

Some of the proteins expressed in protein bands between 48 and 59 kDa probably correspond to functional units or polypeptides of *N*-glycosylated *C. aspersum* hemocyanin resulting from proteolytic processes. The presence of different forms of hemocyanin in the mucus of land snails has been reported in several studies in recent years [23,31,32].

The protein at 84–85 kDa probably corresponded to epiphragmin, identified in the adhesive mucus secretion of *H. aspersa, H. pomatia*, and *Cernuella virgata* land snails [42,43,44,45].

The electrophoretic analysis also showed proteins with an MW above 100 kDa, which most likely represent different types of mucins, proteoglycans, and collagen, which were confirmed in recent studies on the mucus proteins of *C. aspersum* [34,43], *A. fulica* [31,32], and *H. lucorum* [29] land snails through integrative “omics” approach. 

#### 2.2.4. Characterization of the Mucus Fraction with an MW > 20 kDa by Tandem Mass Spectrometrical Analyses of 12% SDS-PAGE

In order to identify some of these hypothesized proteins, the protein bands from 12% SDS-PAGE were excised from the gel. After trypsin digestion, the extracted peptides were analyzed via tandem mass spectrometry and bioinformatics. The amino acid sequences (AASs) of extracted peptides from each protein band were determined by MS/MS analyses, because the identification of proteins by only a Mascot search of the experimental peptide masses of [M+H]^+^ did not lead to satisfactory results. 

Limited proteomic information on gastropods in the NCBI database probably caused this unsuccessful protein identification. In Figure 6, the MALDI-MS/MS spectrum of peptide [M+H]^+^ at *m*/*z* 1670.93 extracted from the protein band at 48.75 ± 1.5 kDa is presented. 

The determined amino acid sequence HGSPIGVPYWDWTR shows 100% identity (E = 2 × 10^−9^) with hemocyanin alpha-N from *C. aspersum* (AYO86684.1). 

In this way, the rest of the proteins were also determined (Table 2). Most of the presented hits demonstrate identities above 60% and E-values between 1 × 10^−13^ and 2 (Table 2; Appendix A). It is known that the lower the E-value, the more “significant” the match is, suggesting a higher probability that the sequences share a common evolutionary origin.

The results shown in Table 2 confirm the presence of mucus protein [QEG59314, from *C. aspersum*]; NADH dehydrogenase subunit 6 [*Albinaria caerulea*]; glutathione S-transferase omega-1 [*P. canaliculata*]; H-type lectins, including *H. aspersa* agglutinin; mucus protein with an MW of 39.115 kDa [QEG59312 from *C. aspersum*], which is probably homologous with von Willebrand factor A domain-containing protein 3B; functional unit β_C_-d of *C. aspersum* hemocyanin; L-amino-acid oxidase-like protein (as Achacin from *L. fulica*); FMRFamide-activated amiloride-sensitive sodium channel; zinc finger protein; elastin-like protein; several types of collagen (collagen alpha-1, collagen α-4, and collagen alpha-6); and mucins (mucin-5AC-, mucin-5B-, mucin-2-, and mucin-17-like proteins). Most of the detected proteins can be associated with antimicrobial and antioxidant properties of this protein mucus fraction. 

### 2.3. Antimicrobial Effect of Two Fractions Isolated from C. aspersum Mucus

The antibacterial activity of both mucus extract fractions with an MW < 20 kDa and an MW > 20 kDa from *C. aspersum* snails was determined against different bacterial strains via a minimum inhibitory concentration (MIC) analysis. Two strains of Gram-positive pathogenic bacterial isolates (*Bacillus cereus* 1085, *Propionibacterium acnes* 1897) and three strains of Gram-negative ones (*Salmonella enterica* 8691, *Enterococcus faecalis* 3915, and *Enterococcus faecium* 8754) were used as test microorganisms. The obtained results showed a different degree of effectiveness of the investigated substances. 

The MIC values presented in Figure 7(a1–b2) show a higher antimicrobial potential of the fraction with an MW > 20 kDa of *C. aspersum* mucus compared to the fraction with an MW < 20 kDa, inhibiting at concentrations of 32–128 µg/mL more than 90% of the microbial growth of Gram-positive pathogenic bacterial isolates *Bacillus cereus* 1085 and *Propionibacterium acnes* 1897. Furthermore, the effect against Gram-positive microorganisms of the bioactive compounds in this fraction was observed even at low concentrations (2–4 µg/mL), having values higher than the IC_50_.

The antimicrobial activity of the mucus fraction with an MW > 20 kDa against Gram-negative bacteria—*S. enterica* 8691, *E. faecalis* 3915, and *E. faecium* 8754—is also well expressed. Even at concentrations of 8 µg/mL, more than 70% inhibition of bacterial growth was achieved (Figure 8a,b). An analysis of the peptide fraction with an MW < 20 kDa from *C. aspersum* mucus also showed the presence of a well-expressed antibacterial effect (Figure 8a). Still, the reported IC_50_ values were lower than those of the protein fraction with an MW > 20 kDa. They range between IC_50_ and IC_80_ against Gram-positive bacteria and have a maximum IC_70_ against Gram-negative strains at the highest mucus concentrations tested. The weakest effect was reported for *S. enterica* 8691. Interestingly, the antimicrobial activity of both investigated fractions of *C. aspersum* mucus is comparable to that of the antibiotic vancomycin, but only when higher concentrations are used. When peptide and protein mucus fractions are added to the culture medium in concentrations of 8 μg/mL, their effect is greatly reduced. 

### 2.4. Studying the In Vitro and In Vivo Effects of Snail Mucus on Eukaryotic Cells

The yeast *S. cerevisiae* has been used as a model system for eukaryotic cell biology to assess the effect of both isolated bioactive fractions of *C. aspersum* mucus on eukaryotic cell metabolism. A time–kill kinetics test was performed, which revealed that the viability of mucus-treated *S. cerevisiae* cells was not significantly different from the not-treated ones (Figure 9). 

Even at 24 h of cultivation, some growth stimulation effect has been observed, showing a 3–30% increase in the cellular biomass when snail mucus has been added to nutrient media. Moreover, the observed improvement in cell growth did not depend significantly on either the added snail mucus concentration or the bioactive fraction type–with an MW < 20 kDa or with an MW > 20 kDa.

Next, the possible cytotoxic effect of the isolated bioactive snail mucus fractions on eukaryotic cells has been studied. The *S. cerevisiae* cells were treated with concentrations of both fractions exceeding those used in characterizing their antibacterial activity. Then, the alternation in the levels of intracellularly generated ROS and carbonylated proteins was evaluated (Figure 10). It was revealed that in the cells treated with 256 μg/mL snail mucus with either an MW < 20 kDa or with an MW > 20 kDa, ROS generation was decreased by 29% to 38%, respectively, compared to the untreated cells (Figure 10a). Simultaneously, the intracellular levels of oxidized proteins were not significantly different between treated and untreated cells, ranging between 18.37 ± 0.98 and 19.06 ± 0.87 μmol/mg of protein (Figure 10b).

To evaluate the possible mechanism involved in the growth-stimulating and cytoprotective effect of *C. aspersum* snail extracts on *S. cerevisiae* eukaryotic cells, intracellular GSH levels and TAC were evaluated (Figure 11). It was shown that exposure to both mucus fractions with an MW < 20 kDa and an MW > 20 kDa leads to an increase in the cytoplasmic GSH level of 35% and 25%, respectively (Figure 11a). The same tendency was observed when TAC was assessed. The exposure of *S. cerevisiae* cells to the action of snail BACs causes an increase in the total antioxidant capacity of the cell (Figure 11b).

## 3. Discussion

The antibacterial active substances from natural sources are promising for obtaining a new type of antimicrobial agent with reduced or even absent toxicity for humans, but highly effective against pathogenic microorganisms [20]. Most gastropods have high potential as sources of antimicrobial peptides and proteins. Several studies have demonstrated the presence of highly effective peptides and proteins in the mucus of terrestrial snails [21,22,23,24,25,26,32], as well as freshwater [43,46] and marine snails [19,20,47]. The mucus secretions of snails and slugs play a pivotal role in the survival of these organisms in interaction with the environment. It is crucial to continue identifying the natural antimicrobial potential of snail mucus because the rise in drug-resistant bacterial species continues to develop. To date, there are still few studies on the antimicrobial properties of the mucus of terrestrial molluscs, and most of them have focused on the land snail *A. fulica* [26].

The data obtained in the present study showed a strong antibacterial effect of the two mucus extract fractions from *C. aspersum* snails against two Gram-positive pathogenic bacterial stains (*B. cereus* 1085 and *P. acnes* 1897) and three strains of Gram-negative ones (*S. enterica* 8691, *E. faecalis* 3915, and *E. faecium* 8754). The results of the in vitro study showed that high concentrations (128 μg/mL and 64 μg/mL) of both fractions with an MW < 20 kDa and an MW > 20 kDa from the native mucus extract exhibited antimicrobial activity against tested pathogenic bacterial strains comparable to antibiotic VCM. 

The obtained data are in accordance with the results of other studies, which prove the presence of specific oligo- and polypeptides with antibacterial action in the mucus of land snails [21,22,23,25,26,39]. Our previous studies demonstrate that the fraction with an MW < 20 kDa shows strong antibacterial activity against *P. aureofaciens* AP9 in deep inoculation of the bacterium, while the fraction with an MW between 10 and 30 kDa exhibits the highest antibacterial activity using surface inoculation of the bacterial strain *E. coli* NBIMCC 8785 [21]. The mucus extract fraction with an MW > 20 kDa was also found to be most effective against the bacterial strain *C. perfringens* in deep anaerobic cultivation [21]. The observed differences in the antimicrobial activities against the tested pathogens are due to the difference in the active components of the two fractions. 

The characterization of both fractions from the native mucus extract showed that their antimicrobial properties are due to different biocomponents, mainly antimicrobial peptides and polypeptides in the fraction with an MW < 20 kDa and proteins as well as glycoproteins with potential antibacterial properties in the fraction with an MW > 20 kDa. 

The peptide fraction with an MW < 20 kDa from *C. aspersum* mucus contains mostly various peptides < 3 kD with antimicrobial and antioxidant properties, as well as some metabolites such as amino acids, allantoin, glycolic acid, γGSH, and others [48]. In the current study, 16 newly antimicrobial peptides (shown in Table 1) were identified, as were several peptides with higher molecular weights between 3 and 10 kDa and some proteins detected as [M+H]^+^ at *m*/*z* between 10 and 20 kDa. The identified peptides (Table 1) contain high levels of glycine, leucine, valine, and proline amino acid residues, as well as the presence of tryptophan, aspartic acid, phenylalanine, and arginine, which are associated with their antimicrobial activity and antioxidant properties. An amino acid sequence alignment comparison of *C. aspersum* mucus peptides by CAMPSing software (http://www.campsign.bicnirrh.res.in/, accessed on 24 March 2024) revealed high identity (more than 70%) with known AMPs (Appendix A). The structural characteristics of the peptides with the highest prognostic antibacterial and antifungal activity (nos. 17, 18, 21, 22, and 23, Table 1) show high levels of glycine, leucine, valine, and proline residues, indicating that they belong to a new class of antimicrobial peptides, rich in Gly and Gly/Leu, which demonstrate broad-spectrum antibacterial activity [49].

Previous studies have also shown that the presence of peptides with similar amino acid sequences revealed homology with mostly leptoglycin, acanthoscurrin, and ctenidin [21,38,39,40]. In addition, most peptides with high predictive therapeutic values are cationic and/or neutral (with one or two positively charged residues that are neutralized with negatively charged ones). Most of the identified peptides (Table 1) are hydrophobic, which is a known prerequisite for their antimicrobial activity because hydrophobicity not only modulates the peptide–membrane interaction but also affects cellular selectivity and is positively correlated with cytotoxic activity [46,50]. AMPs are known to kill microbes through mechanisms that primarily involve interactions between the charged amino acid residues of the peptide and components of the bacterial membranes. There are several mechanism models with which to explain the actions of these peptides, including the toroidal pore model, the barrel model, and membrane interaction according to the Shai–Huang–Matsazuki carpet model [50,51,52,53]. These interactions can lead to a number of effects, including membrane permeabilization, depolarization, leakage, or lysis, leading to cell death. 

In a fraction with an MW < 20 kDa, some peptides with higher molecular masses between 10 and 20 kDa were also identified. They probably contribute to the antibacterial activity against the tested bacterial pathogens, because they are in accordance with known proteins and glycoproteins with antibacterial activity in the snail mucus of *C. aspersum* [26,35], *Helix pomatia* [36], and *A. fulica* [37]. For example, the polypeptide determined at 12.464 kDa (Figure 2) is probably a lectin previously found in a study by Pietrzyk et al., in the mucus of *C. aspersum* [35]. A similar protein with biological activity against *Streptococcus mutans* and *Aggregatibacter actinomycetemcomitans* was also found in *A. fulica* mucus with 11.45 kDa in [54]. The protein detected by us at *m*/*z* 18.005 kDa in a fraction with an MW < 20 kDa using a MALDI-MS analysis (Figure 2) probably corresponds to the anti-*Pseudomonas aeruginosa* protein (MW of 17.5 kDa) described in a study by Pitt et al. (2019) [26]. 

Several studies have shown that mucus from *C. aspersum* snails also demonstrates antimicrobial activity due to one or more proteins with molecular masses between 30 and 100 kDa [21,22,26,54]. Based on the results of the electrophoretic analysis and proteomic analysis on SDS-PAGE of a mucus fraction with an MW > 20 kDa, high homology was found with some proteins and glycoproteins (*N*-glycosylated proteins, lectins, and mucins) with potential antibacterial activity. Surprisingly, a protein at 17.5 kDa was detected upon an SDS-PAGE analysis of the fraction with an MW > 20 kDa. The alignment of AASs obtained by MALDI-TOF-MS/MS showed homology to mucus protein from *C. aspersum* snails (ID QEG59314.1), identified by Pitt et al., as a lectin with potential antimicrobial activity against strains of *Ps. aeruginosa*. Some lectins have pathogen-recognition receptor properties, which may contribute to their antimicrobial activity. Natural lectins have powerful antibacterial properties because they bind to carbohydrates on microbial surfaces [55]. 

The results of the proteomic analysis show that peptides extracted from the protein band at 26 kDa are homologous on H-type lectin from *B. glabrata* and agglutinin from *H. aspersa* (HAA) as well as *H. pomatia* (HPA), which also play a role in the defense of the snail against pathogens. H-type lectins represent a new group of lectins that have been identified in invertebrates. In snails, H-type lectins (e.g., HPA, HAA) protect fertilized eggs from bacteria because they are secreted by snail protein glands as a component of the perivitelline fluid [56,57]. Lectins are involved in many biological processes, including cell recognition, viral and bacterial pathogenesis, and inflammation [56]. Some studies have shown that high-molecular-weight lectin isolated from the mucus of the giant African snail *A. fulica* induces agglutination of Gram-positive and Gram-negative bacteria, although it does not inhibit bacterial growth [58]. In addition, H-type lectins recognize glycans present in the cell wall of pathogenic bacteria; for example, group C *Streptococcus* sp. Agglutinins are glycoproteins in nature and play a fundamental role in the innate immune responses in invertebrates [59]. 

The AASs of several peptides extracted from the protein band at 39.115 kDa (Table 2) show high homology with antibacterial mucus protein with an MW of ~37.5 kDa (QEG59312.1, *C. aspersum*, named “Aspernin”), active against different strains of *Ps. aeruginosa* [26]. According to Pitt et al., the protein “Aspernin”, which is probably glycosylated, shares homology with proteins in another species of snail—*Biomphalaria glabrata*—such as H-type lectin domain-containing protein [26]. Our results also showed that some peptides derived from this protein band matched domains of the von Willebrand factor A superfamily. The proteins with an MW between 30 and 40 kDa found in the fraction with an MW > 20 kDa are in full agreement with the recently detected proteins by Pitt et al., which suggests that these proteins are lectins and are also related to the antimicrobial properties of the mucus [26].

Another identified protein in the mucus fraction with an MW > 20 kDa with antibacterial activity is hemocyanin. Our results (presented in Figure 5 and Table 2) revealed that the intensive band in the range of 48.75 ± 1.5 kDa to 97.0 ± 1 kDa is composed of proteins demonstrating high homology with hemocyanin functional units and subunits. The high protein expression of different forms of hemocyanin can explain observed antibacterial activity against tested bacterial pathogens (Figure 8(b1,b2) and Figure 9(b1–b3)). Previous studies have also demonstrated that molluscan hemocyanin possesses antibiotic activity [60,61,62], which supports our hypothesis. Interest in hemocyanin as a bioactive molecule against pathogenic microorganisms is relatively recent [60,63,64]. Information about the antibiofilm and antiviral activity of hemocyanin from land snails is still very scarce [23]. Moreover, hemocyanin-derived polypeptides (detected at 40, 60, and 80 kDa) with possible antimicrobial properties were previously reported by Suárez et al. (2021) [23]. Furthermore, Dolashka et al. (2016) reported the antimicrobial activity of the βc-HaH structural subunit of *H. aspersa* (also named *C. aspersum*), which demonstrated strong selective antimicrobial activity against *S. aureus*, *Streptococcus epidermidis*, and *Escherichia coli*, while *Ps. aeruginosa* and *E. faecium* developed unconditionally with its presence [18]. Therefore, the observed antibacterial activity of fractions with an MW > 20 kDa against *B. cereus* 1085, *P. acnes* 1897, *S. enterica* 8691, *E. faecalis* 3915, and *E. faecium* 8754 cannot be fully explained only by the identification of hemocyanin. 

In addition, on the protein band at 52.89 ± 1.0 kDa, another protein was found to be homological with the *C. aspersum* mucus protein (QEG59312.1) and an unnamed protein from *Candidula unifasciata* snails (CAG5131631.1 and CAG5131621.1). This protein is likely related to the antimicrobial activity of the fraction with an MW > 20 kDa. The study by Pitt et al. [26] also mentioned a glycoprotein with an MW at ~50 kDa with potential antimicrobial activity against *Ps. aeruginosa*. Furthermore, we found that the protein detected at the range 56.94–59.04 kDa shared homology with proteins with L-amino acid oxidase activity, such as achacin, which is established as an antibacterial *N*-glycosylated protein in the mucus of *L. fulica* and *A. fulica* snails. Several studies have associated the broad spectrum of the antimicrobial properties of the mucus of the giant African snail *A. fulica* against Gram-positive bacteria (*Bacillus subtilis* and *Staphylococcus aureus*) and Gram-negative strains (*Escherichia coli* and *Ps. aeruginosa*) mainly with the presence of achacin. 

In another study [23] it was shown that the activity against the biofilm-forming bacteria *S. aureus* was found to be mainly due to the action of two glycoproteins—hemocyanin and “achacin”, as well as their fragments resulting from proteolysis processes. Indeed, *N*-glycosylated achacin, in addition to inhibiting bacterial growth, also appears to attack bacterial plasma membranes [33]. It was found that the antibacterial activity of achacin depends on the production of H_2_O_2_, which is obtained from the oxidative deamination reaction. These data indicate that achacin can also attack pathogens during other growth phases by increasing the local H_2_O_2_ concentration to not harm neighboring host cells [32,65].

Extracellular matrix proteins constitute 40–50% of the mucus proteins of *C. aspersum* [30]. Our results have proven the presence of homologous proteins on mucin-5AC-, mucin-5B-, mucin-2-, and mucin-17-like proteins, as well as collagens. Mucins, included in snail mucus, can play roles for several biological functions, including adhesion, lubrication, and microbial protection [66,67]. Mucins are among the largest macromolecules, characterized by a protein core and branching glycan chain. The structural diversity of mucins allows for their extensive biological diversity and unique physical characteristics. A tandem repeat domain located in the center of the protein backbone, rich in serine, threonine, and proline, serves as an anchor for glycosylation. Mucin glycans are predominantly *O*-linked, but minor amounts of *N*-linked glycans can also be present [66,68]. It appears that mucins, which are an important component of snail mucus, play an important role as the first line of defense against bacterial infections; however, how the glycoprotein mucin—which makes up the mucus—interacts with the bacteria is still not fully understood [69]. For example, the MUC2 mucin established in the mucus fraction, but also found in the human colon, is capable of aggregating Gram-protein bacteria such as *Bacillus subtilis* and *E. faecalis* by binding to the peptidoglycan present in the cell wall, thereby inhibiting epithelial cell penetration. Furthermore, both mucins MUC5AC and MUC5B have been shown to inhibit up to 85% of the *Pseudomonas aeruginosa* isolated from the sputum of patients with a chronic infection [69]. The antimicrobial potentials of mucus mucins from different species of giant African land snails on some typed culture pathogenic bacteria were studied by Okeniyi et al. (2022) [70]. It was found that the snail mucus extract had antimicrobial effects on Gram-positive and Gram-negative bacteria, but mucus mucins seem to lose their antibacterial potential with time [70]. *A. marginata*’s mucus secretions had stronger antibacterial activity against *B. subtilis* when compared to mucus from *A. achatina* and *A. fulica* [24]. 

Several studies show that polysaccharides such as glycosaminoglycans (GAGs) were also found in the mucus of land snails [30,31,71,72,73]. Deng et al. (2023) showed that sulfated glycosaminoglycan in the mucus of *A. fulica* and *H. lucorum* effectively promoted chronic wound healing in a diabetic rat model by improving skin incision adhesion as well as accelerating granulation tissue regeneration, angiogenesis, and collagen deposition [31]. Recently, Kodchakorn et al. (2023) reported that heparin-like sulfated glycosaminoglycan (named acharan sulfate) found in the mucus of *A. fulica* has promising potential for the preparation of sulfated polysaccharides as an alternative to heparin and shows antiviral properties [71]; however, there is still no information about the antibacterial action of the mucus glycosaminoglycans. 

We hypothesize that the synergy between the bioactive components, their composition determined mainly by proteins and glycoproteins, such as a mucus protein named aspernin, *N*-glycosylated *C. aspersum* hemocyanin, lectins, L-amino acid oxidase-like protein, and mucins, is responsible for the high antibacterial effect of the mucus fraction > 20 kDa compared to vancomycin against tested bacterial pathogens. 

Moreover, the performed study demonstrates that both isolated fractions from the mucus extract of *C. aspersum* with an MW < 20 kD and an MW > 20 kDa are not cytotoxic to tested eukaryotic cells of *S. cerevisiae*, but are characterized by growth-promoting potential. Bearing in mind that microorganisms and various Mollusca frequently form symbiotic relationships that are crucial to the ecology and evolution of species [74], it could be suggested that mucus from *C. aspersum* is used by *S. cerevisiae* as a supplementary substrate for its growth. Adding both mucus fractions to the nutrient media further leads to a substantial reduction in ROS levels. ROS generated during cell growth are involved in the roughly 60 distinct pathways that lead to protein oxidation, which includes carbonylation causing an unbalanced cellular proteome and malfunctions [75]. The obtained results suggested that *C. aspersum* mucus possesses ROS scavenging properties, which automatically reflect their ability to reduce the level of carbonylated proteins in eukaryotic organisms while combating pathogenic bacterial strains. Thus, the two isolated bioactive fractions from *C. aspersum* can not only be used effectively to treat antibiotic-resistant pathogens, but their administration also has additional beneficial effects related to the reduction in oxidative stress in a host organism’s cells. In fact, other authors confirmed the radical scavenging activity of snail mucus, showing that *P. canaliculata* and *L. fulica* mucus have pro-oxidant capacity [76]. Furthermore, the same authors observed that *L. fulica* mucus exhibited higher antioxidant capacity, similar to those we found for the *C. aspersum* low- and high-molecular-weight mucus fractions. According to Wang et al. (2010) [77], low-molecular-weight components found in snail mucus, including uronic acid, phenolic compounds, vitamins C and E, and uric acid, determine their antioxidant qualities. 

The specific characteristics of the many mucus peptides, such as their hydrophobicity, molecular mass, and AASs, are also related to their antioxidant capacity [78]. The determined composition of the fraction with an MW > 20 kDa is consistent with its antioxidant and cytoprotective properties, which are probably due to the complex synergistic actions of the specific compounds found in it. Several studies in recent years have also confirmed the antioxidant properties of *C. aspersum* mucus [27,28,29].

## 4. Materials and Methods

### 4.1. Preparation of Mucus Extract

The mucus was collected from *C. aspersa* snails, grown on Bulgarian eco-farms by patented technology without disturbing their biological functions [21,79]. The method for collecting native extracts from garden snails based on electrical stimulation with a low voltage was used and can be used repeatedly to extract mucus. Thus, obtained crude mucus extracts were homogenized and centrifugated to remove coarse impurities. The supernatant was subjected to several cycles of filtration, using filters of decreasing pore size for each subsequent filtration. The protein concentration in the native mucus extract was determined by a Bradford assay [80]. The obtained crude mucus extract was divided into two fractions by ultrafiltration, using membranes with pore sizes of 20 kDa (polyethersulfone, Microdyn Nadir™ from STERLITECH Corporation, Go-leta, CA, USA, respectively): a peptide fraction with an MW below 20 kDa and a fraction containing compounds with an MW above 20 kDa.

The peptide fraction with an MW < 20 kDa was additionally separated into three fractions using Amicon^®^ Ultra-15 centrifugal tube filters with 3 and 10 kDa membranes. Finally, the following samples were obtained: sample 1—a fraction with compounds with an MW < 3 kDa; sample 2—a fraction with compounds with an MW of 3–10 kDa; and sample 3—a fraction with compounds with an MW of 10–200 kDa.

The use of a noninvasive technique—ultrafiltration—ensured that we obtained fractions containing intact compounds.

### 4.2. HPLC Purification of Peptides from the Fraction below 20 kDa

Reverse-phase high-performance liquid chromatography (RP-HPLC) on a ZURA^®^ HPLC system (KNAUER, Berlin, Germany) was used to purify the mucus fraction with an MW below 20 kDa after lyophilization on a BioSill C18 HL 90–10 column (250 mm × 10 mm, Bio-Rad Life Science Group, 1000 Alfred Nobel Drive, Hercules, CA 94547, USA) equilibrated with 0.1% trifluoroacetic acid (TFA, *v*/*v*) (solution A). Elution was performed by a gradient of water–acetonitrile formed by solution A (0.1% TFA/water) and solution B (100% acetonitrile in 0.1% TFA (*v*/*v*)) at a flow rate of 1.0 mL/min, for 70 min, as follows: to 10 min., A 100%; 11–20 min, A 90%; 21–50 min, A 40%; 51–60 min, A 0%; 61–63 min, A 0%; and 64–70 min, A 100%. Ultraviolet absorption was monitored at 216 nm. 

The eluted fractions were collected and dried via vacuum concentration with a Speed-110 vac. The fractions were reconstituted in Milli Q water containing 0.10% TFA (*v*/*v*).

### 4.3. Molecular Mass Analysis and De Novo Sequencing of Peptides by Mass Spectrometry

The isolated peptide fraction with an MW < 3 kDa (sample 1) was lyophilized and analyzed via MALDI-TOF-TOF mass spectrometry on an AutoflexTM III High-Performance MALDI-TOF and TOF/TOF System (Bruker Daltonics, Bremen, Germany), which uses a 200 Hz frequency-tripled Nd–YAG laser operating at a wavelength of 355 nm. An analysis was carried out after mixing 2.0 μL of the sample with 2.0 μL of matrix solution (7 mg/mL of α-cyano-4-hydroxycinnamic acid (CHCA) in 50% ACN containing 0.1% TFA), but only 1.0 μL of the mixture was spotted on a stainless steel 192-well target plate. The samples were allowed to dry at room temperature before being analyzed. A total of 3500 shots were acquired in the MS mode, and a collision energy of 4200 was applied. A mixture of angiotensin I, Glu-1-fibrinopeptide B, ACTH (1–17), and ACTH was used for the calibration of the mass spectrometer. 

The MS/MS spectra were carried out in reflector mode with external calibration using fragments of Glu-fibrino-peptide B. The amino acid sequences of peptides were identified via precursor ion fragmentation using MALDI-MS/MS analysis.

### 4.4. Sodium Dodecyl Sulfate Polyacrylamide Gel Electrophoresis (SDS-PAGE)

The eluted intravenous immunoglobulins were analyzed by one-dimensional sodium dodecyl sulfate-polyacrylamide gel electrophoresis (1D-SDS-PAGE) using 5% stacking gel and 12% resolving gel, according to the Laemmli method with modifications [81]. The samples were dissolved in a Laemmli sample buffer containing 10 mM DTT as the reducing agent. A staining Coomassie Brilliant Blue G-250 buffer was used for the visualization of proteins on the gel, as was a protein standard marker–mixture of proteins with molecular weights from 6.5 kDa to 200 kDa (SigmaMarkerTM, Sigma-Aldrich, Saint Louis, MO, USA).

### 4.5. Image Analysis of 10% SDS-PAGE by ImageQuant™ TL v8.2.0 Software

The obtained polyacrylamide gel (PAG) was captured on an Image Scanner III (GE Healthcare), and the image was opened within the ‘1D gel analysis’ utility of ImageQuant TL v8.2 software (GE Healthcare Bio-Sciences AB, Uppsala, Sweden), which is highly automated and easy-to-use image analysis software. The background was performed through the “image rectangle” setting to compensate for the intensity of the image background. All bands were identified manually, including those in the standard protein marker, with a pen tool, as the bandwidth was fixed at 54 pixels. 

The molecular weight analysis of each band was performed using the protein data standard SigmaMarker^TM^ (Sigma-Aldrich, Saint Louis, MO, USA) in the range of 6500–200,000 Da. Automatically, horizontal bands were drawn to the individual bands of the MW marker and calculated with the cubic curve spline. 

Based on the precalculated amount of bands at the marker, the amount of bands tested is determined [82].

### 4.6. Tryptic In-Gel Digestion and Peptide Extraction

Protease digestion was run according to the work of Rosenfeld et al. with a slight modification [20]. The target protein bands excised from the SDS-PAGE gels were washed twice with a 150 µL mixture of 50% acetonitrile (ACN) and 200 mM NH_4_HCO_3_ each for 20 min at 30 °C to decolorize, as described previously in [83]. The digestion of proteins in gel was carried out with porcine trypsin (Promega, Madison, WI, USA). After drying the decolorized gels in a speed vac concentrator, a volume of 10 μL of digestion buffer (50 mM ammonium bicarbonate, pH 7.8, containing modified trypsin) was added to them, and the Eppendorf tubes were kept on ice for 45 min to allow the gel pieces to be completely soaked with the protease solution. 

Digestion was performed overnight at 37 °C, the supernatants were recovered, and the resulting peptides were extracted twice with 35 μL of 60% ACN/0.1% HCOOH. The extracts were pooled and dried in the speed vac concentrator.

### 4.7. Antibacterial Activity Assay

The antibacterial activity of the two bioactive fractions from *C. aspersum* was tested against 2 Gram-positive and 3 Gram-negative pathogenic bacteria, listed below. The bacterial strains *Bacillus cereus* NBIMCC 1085, *Propionibacterium acnes* DSM 1897, *Salmonella enterica* NBIMCC 8691, *Enterococcus faecalis* NBIMCC 3915, and *Enterococcus faecium* NBIMMCC 8754 were obtained from the National Bank for Industrial Microorganisms and Cell Cultures (Sofia, Bulgaria). It has been determined according to the procedure for the minimum inhibitory concentrations (MICs) using the broth microdilution method according to Clinical Laboratory and Standards (CLSI) guidelines [84]. Briefly, the bacterial suspension cultured to the logarithmic phase was diluted to a 0.5 McFarland standard (approximately 1.5 × 10^8^ CFU/mL) and then diluted 150 times to 1 × 10^6^ CFU/mL using nutrient media. A 50 μL volume of undiluted and serial twofold dilutions of BACs with Mueller Hinton Broth (and Brain Heart Infusion Broth for *P. acnes*) was dispensed in 96-well microtiter plates. Subsequently, an equal volume of adjusted inoculum (1 × 10^6^ CFU/mL) was added to each well of the microtiter plates up to a final volume of 100 μL. The nutrient media with a bacterial culture without bioactive fractions were used as a negative control, and in the positive control bioactive fractions were replaced by the glycopeptide antibiotic vancomycin (VCM) (stock solution 40 mg/mL). The MIC value is accepted as the lowest concentration of snail BACs at which bacterial growth is completely inhibited.


**Bacteria Strain**

**Causative Agents**
*Bacillus cereus* NBIMCC 1085Foodborne disease*Propionibacterium acnes* DSM 1897Skin condition of acne, chronic endophthalmitis, corneal ulcers, sarcoidosis, etc.*Salmonella enterica* NBIMCC 8691Salmonellosis*Enterococcus faecalis* NBIMCC 3915Endocarditis, sepsis, urinary tract infections (UTIs), meningitis, etc.*Enterococcus faecium* NBIMMCC 8754Bloodstream infections, urinary tract infections (UTIs), and wound infections associated with catheters or surgery are the most common infections

### 4.8. In Vivo and In Vitro Effect on Eukaryotic Cell

The effect of both isolated fractions from the mucus of *C. aspersum* on eukaryotic cells was evaluated using *Saccharomyces cerevisiae* as a model organism.

#### 4.8.1. Cell Treatment and Viability Assay

*Saccharomyces cerevisiae* NBIMCC 584 was obtained from the National Bank for Industrial Microorganisms and Cell Cultures (Sofia, Bulgaria). The cells were cultured in a standard YPD medium containing 2% glucose, 1% yeast extract, and 1% peptone (pH 6.5) at 30 °C. A time–kill kinetics test was performed, adding various concentrations of mucus fractions with an MW < 20 kDa and with an MW > 20 kDa to the growth media (0, 64, 128, and 256 μg/mL). Yeast growth was quantified after incubation for 0, 3, 6, 12, and 24 h. Absorbance was measured at 600 nm using a microplate reader. 

For in vitro experiments, 1 g of *S. cerevisiae* biomass was treated with both mucus fractions at a final concentration of 256 μg/mL for 1 h at 25 °C, washed twice with distilled water and subjected to mechanical disruption according to the procedure described by [85]. Obtained cell-free extracts were further used for biochemical analysis.

#### 4.8.2. Measurement of the Mucus Cytotoxic Effect

The cytotoxic effects of the two mucus extracts were evaluated by measuring the alternations in the intracellular levels of ROS and carbonylated proteins. The generation of ROS in the *S. cerevisiae* cells was examined using the nitroblue tetrazolium test (NBT) method described by Kostova et al. (2008) [86]. The levels of oxidized proteins were assessed following the procedure of Mesquita et al. (2014) [87].

#### 4.8.3. Determination of Cytoprotective Effect

The reduced glutathione (GSH) levels in the cells were measured using Zhang’s method (2000) [88]. The total antioxidant capacity (TAC) has been evaluated using Kumaran and Karunakaran’s (2006) phosphomolybdenum method [89]. 

## 5. Conclusions

In summary, the study reveals, for the first time, promising antibacterial potential against various pathogenic bacterial strains of both fractions from *C. aspersa* mucus. The fraction with an MW > 20 kDa showed higher antimicrobial activity against *B. cereus* 1085, *P. acnes* 1897, *S. enterica* 8691, *E. faecalis* 3915, and *E. faecium* 8754 compared to vancomycin. Furthermore, both mucus fractions demonstrated noncytotoxic effects on model eukaryotic cells of *S. cerevisiae*. In addition, a positive effect by reducing the levels of intracellular oxidative damage and increasing the antioxidant capacity on *S. cerevisiae* cells were found for both mucus extract fractions of *C. aspersum*. The study showed 16 novel peptides with an MW between 1000–2900 Da identified by de novo sequencing in the mucus fraction with an MW < 20 kDa, with potential antibacterial activity. Some proteins and glycoproteins identified via a proteomic analysis with SDS-PAGE and bioinformatics are responsible for the antibacterial effect of the mucus fraction with an MW > 20 kDa. The results showed high homology with an antibacterial protein named aspernin, H-type lectins, N-glycosylated *C. aspersum* hemocyanin, L-amino acid oxidase-like protein, mucins (mucin-5AC-, mucin-5B-, mucin-2-, and mucin-17-like proteins), and glutathione S-transferase as well as NADH dehydrogenase.

Further studies are needed to reveal the antibacterial potential of both mucus fractions as new alternative therapeutics with a low risk for antimicrobial resistance.

## Figures and Tables

**Figure 1 molecules-29-02886-f001:**
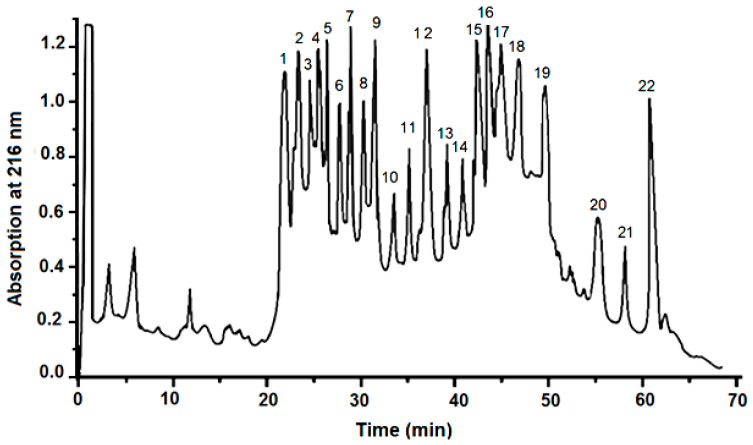
Purification of RP-HPLC of the mucus fraction with an MW < 20 kDa on a BioSill C18 HL 90–10 column (250 mm × 10 mm) via a gradient of water–acetonitrile (with 0.1% TFA) for 70 min at a flow rate of 1.0 mL/min.

**Figure 2 molecules-29-02886-f002:**
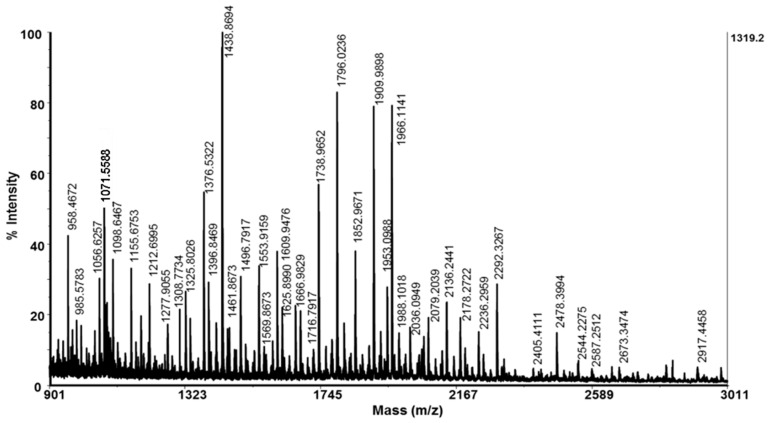
MALDI-Tof-MS spectrum of the fraction with an MW < 3 kDa. A standard peptide solution was used to calibrate the mass scale of an Autoflex™ III High-Performance MALDI-TOF and TOF/TOF system (Bruker Daltonics, Bremen, Germany).

**Figure 3 molecules-29-02886-f003:**
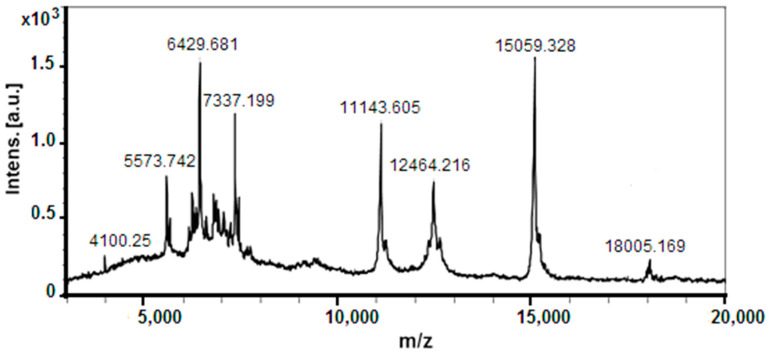
MALDI-Tof-MS spectrum of the fraction with an MW < 20 kDa recorded in the range 3–20 kDa. A standard peptide solution was used to calibrate the mass scale of an Autoflex™ III High-Performance MALDI-TOF and TOF/TOF system (Bruker Daltonics, Bremen, Germany).

**Figure 4 molecules-29-02886-f004:**
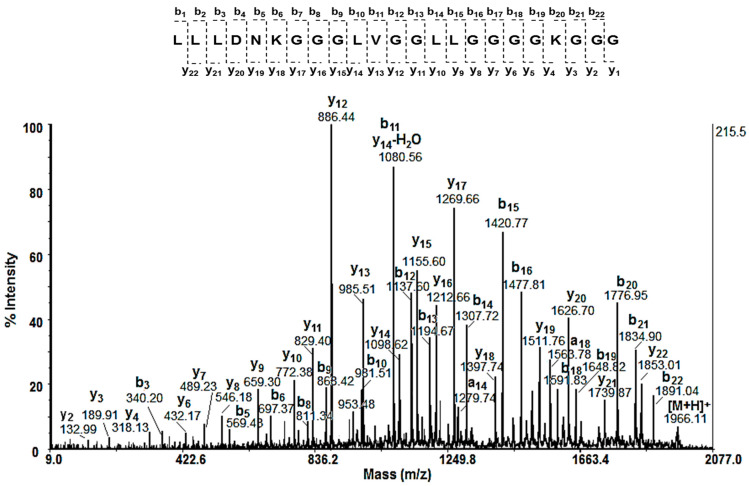
MALDI-MS/MS spectrum of peptide [M+H]^+^ at *m*/*z* 1966.11 Da and its amino acidic sequence LLLDNKGGGLVGGLLGGGGKGGG, determined by de novo sequencing.

**Figure 5 molecules-29-02886-f005:**
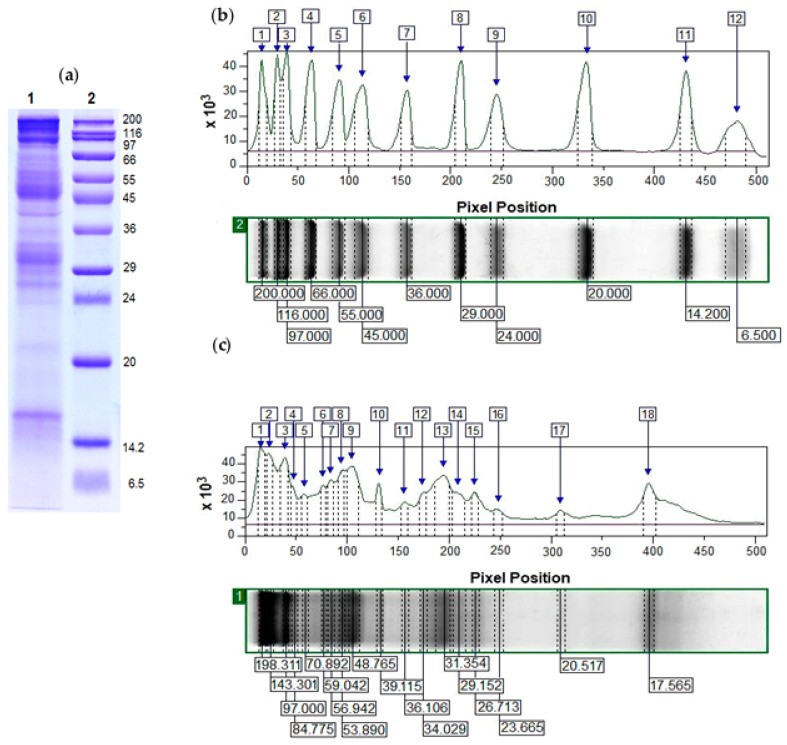
An analysis of molecular masses and protein intensities of 12% SDS-PAGE, scanned with a high resolution, using ImageQuant^TM^ TL v8.2.0 software. (**a**) Electrophoretic pathway: (**1**) standard protein marker with an MW between 200 and 6.5 kDa (SigmaMarker^TM^, Sigma-Aldrich, Saint Louis, MO, USA); (**2**) mucus extract fraction from *C. aspersum* with an MW > 20 kDa. (**b**) Electrophoretic profile of a standard protein molecular marker (electrophoretic lane 2) analyzed by ImageQuant^TM^ TL. (**c**) Analysis of the electrophoretic profile of the mucus extract fraction from *C. aspersum* with an MW > 20 kDa (electrophoretic lane 1), using ImageQuant^TM^ TL.

**Figure 6 molecules-29-02886-f006:**
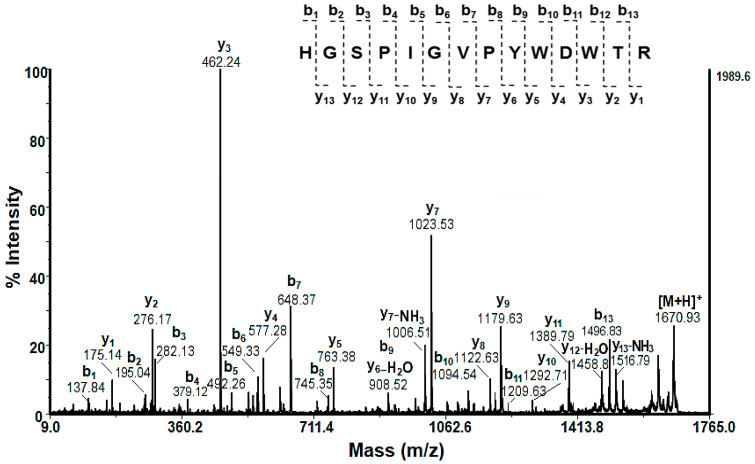
MS/MS spectrum of peptide [M+H]^+^ at *m*/*z* 1670.93 from the protein band at 48.75 kDa of the mucus extract fraction with an MW > 20 of *C. aspersum* mucus.

**Figure 7 molecules-29-02886-f007:**
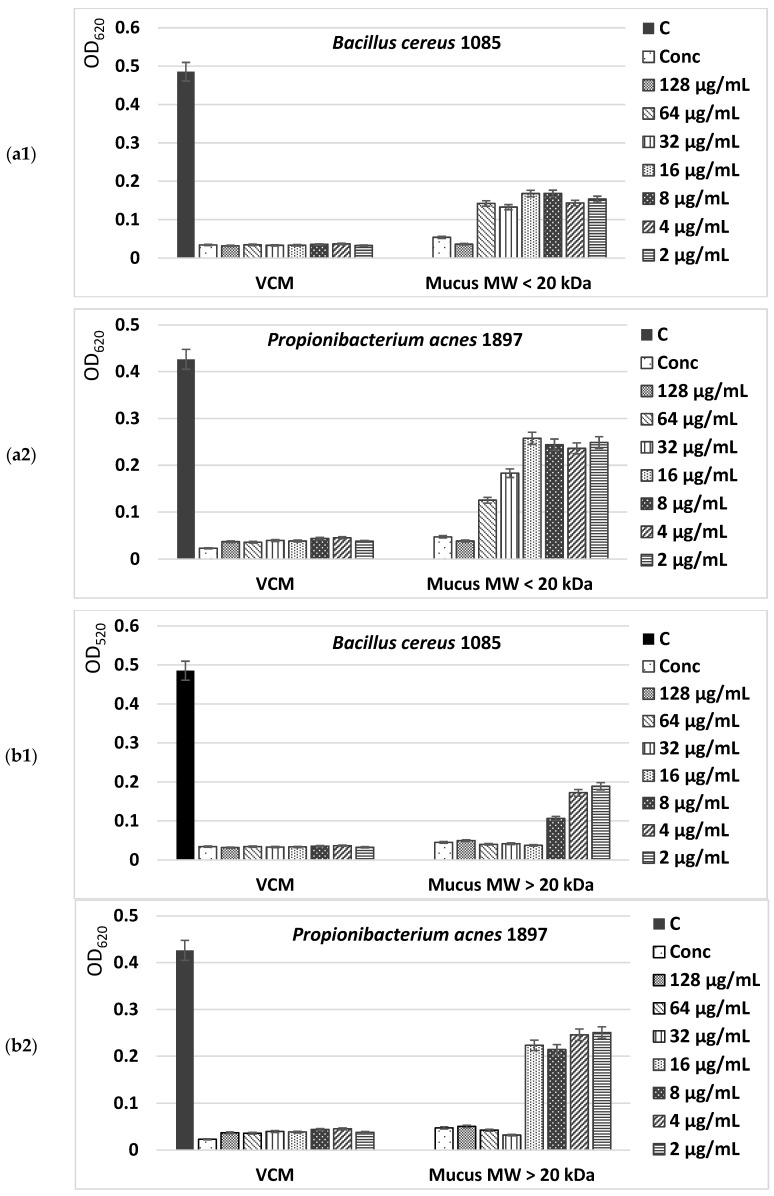
Antibacterial activity of two mucus fractions from *C. aspersum* snails with an MW < 20 kDa (**a1**,**a2**) and an MW > 20 kDa (**b1**,**b2**) against selected Gram-positive pathogenic strains. As a positive control the antibiotic vancomycin (VCM) is used. The control (C) represents untreated cells. Treatments were carried out with concentrated (Conc) and diluted fractions of snail mucus.

**Figure 8 molecules-29-02886-f008:**
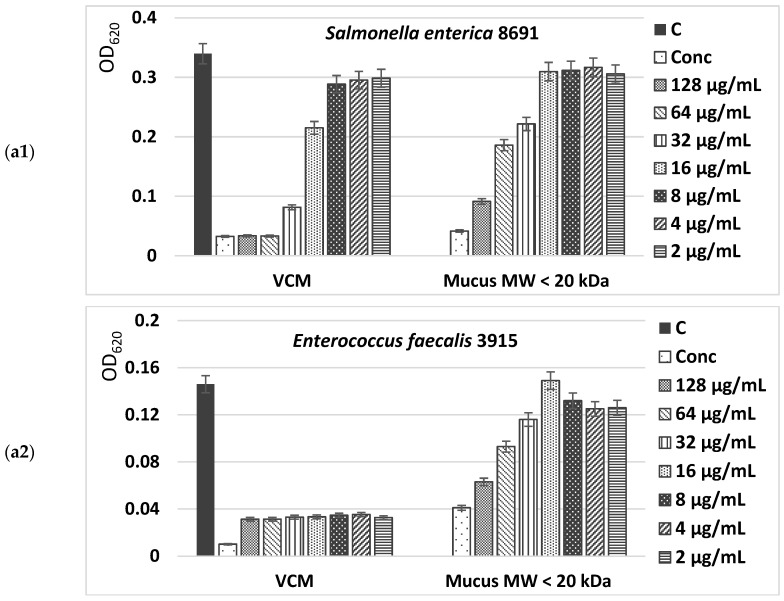
Antibacterial activity of both fractions extracted from the mucus of *C. aspersum* snails with an MW < 20 kDa (**a1**–**a3**) and an MW > 20 kDa (**b1**–**b3**) against selected Gram-negative pathogenic strains. As a positive control, the antibiotic vancomycin (VCM) is used. The control (C) represents untreated cells. Treatments were carried out with concentrated (Conc) and diluted fractions of snail mucus.

**Figure 9 molecules-29-02886-f009:**
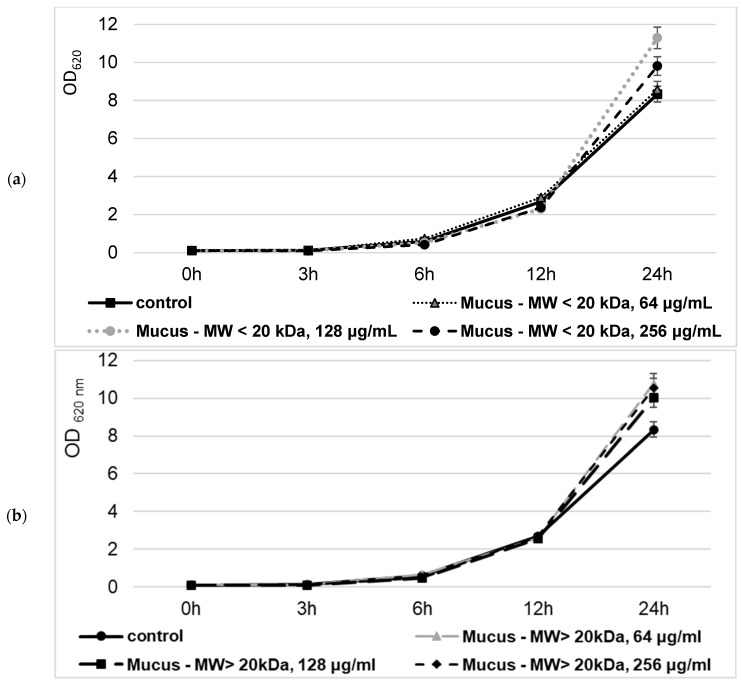
Effect of the *C. aspersum* mucus fraction with an MW < 20 kDa (**a**) and an MW > 20 kDa (**b**) on *S. cerevisiae* NBMCC 584 growth.

**Figure 10 molecules-29-02886-f010:**
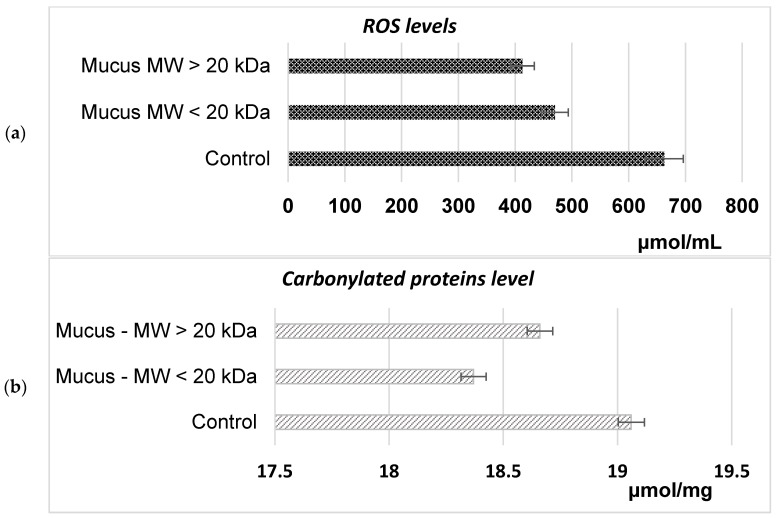
Intracellular ROS (**a**) and carbonylated protein (**b**) levels in *S. cerevisiae* NBMCC 584 cells grown in the presence of *C. aspersum* mucus fractions with an MW < 20 kDa and an MW > 20 kDa.

**Figure 11 molecules-29-02886-f011:**
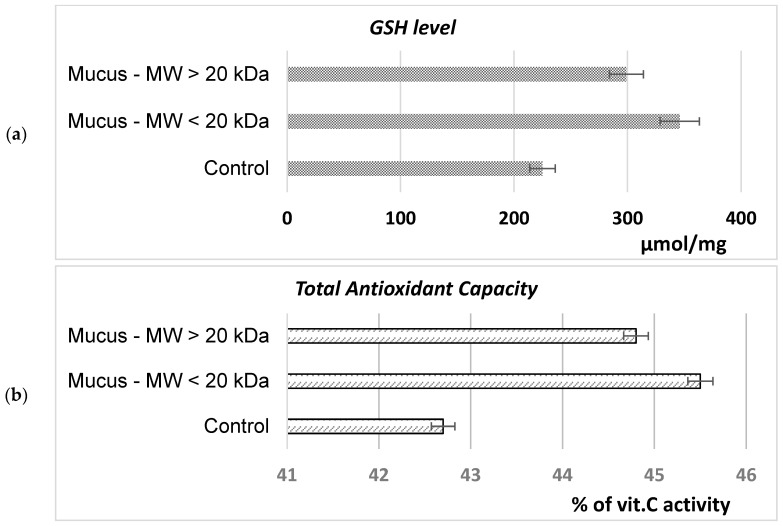
Effect of *C. aspersum* mucus fractions with an MW < 20 kDa and an MW > 20 kDa on *S. cerevisiae* NBMCC 584 cellular reduced glutathione levels (GSH) (**a**) and total antioxidant capacity (TAC) (**b**).

**Table 1 molecules-29-02886-t001:** The primary structures of peptides with an MW below 3 kDa, from a garden snail, *C. aspersum*, identified by de novo sequencing in MALDI-MS/MS.

No	Amino Acid Sequence of Peptides	[M+H]^+^ Da	Calcul. Monois. Mass Da	pI	GRAVY	Net Charge	Predicted by iAMPpred Software http://cabgrid.res.in:8080/amppred (accessed on 25 April 2024)		
							Antibacterial (%)	Antiviral (%)	Antifungal (%)
1	LLPFKEPDL	1071.60	1070.60	4.37	−0.600	−2/+1	28	51	19
2	ACGATLQLENCG	1179.77	1178.51	4.00	+0.350	−1/0	29.3	35.7	43.7
3	LNLGGNGANGLVGG	1212.76	1211.63	5.52	+0.321	0/0	74.0	43.1	74.3
4	AGVGGAAGNPSTYVG	1277.70	1276.60	5.57	+0.260	0/0	25.1	7.4	11.3
5	GGGMVKEDGSCLGV	1308.77	1307.58	4.37	+0.207	−2/+1	40.4	31.4	33.7
6 ^a^	MLGGGVNSLRPPK	1325.80	1324.73	11.0	−0.262	0/+2	22.8	14.0	8.9
7	CVGGAGGHGDSCAKGT	1376.53	1375.56	6.73	−0.106	−1/+1	85.2	48.8	74.4
8	GGGGYHTWGEGGKF	1409.48	1408.62	6.75	−0.964	−1/+1	69.0	62.8	72.6
9	MLNVAVNKGEVKH	1438.86	1437.78	8.37	−0.138	−1/+2	56.4	38.0	19.7
10 ^b^	NLVGGSGGGGRGGANPLG	1496.79	1495.75	9.75	−0.217	0/+1	66.0	33.7	48.2
11	GTMSPAGGEMGPVTAGVG	1576.04	1574.71	4.00	+0.250	−1/0	13.1	24.8	8.3
12	GTKGCGPGSCPPGDTVAGVG	1716.79	1715.76	5.82	−0.100	−1/+1	23.8	20.2	25.8
13 ^c^	ACSLLLGGGGVGGGKGGGGHAG	1738.96	1737.86	8.27	+0.409	0/+1	82.6	49.5	67.0
14 ^a^	LLLDGFGGGLLVEHDPGS	1796.00	1794.92	4.02	+0.439	−3/0	37	45	10
15 ^d^	MGGWGGLGGGHNGGWMPPK	1852.96	1851.83	8.52	−0.611	0/+1	69	56.0	57.0
16 ^c^	ACLTPVDHFFAGMPCGGGP	1876.88	1875.81	5.08	+0.542	−1/0	32.4	42.6	20.1
17 ^c^	NGLFGGLGGGGHGGGGKGPGEGGG	1909.98	1908.88	6.75	−0.487	−1/+1	89.5	67.2	79.5
18	LLLDNKGGGLVGGLLGGGGKGGG	1966.11	1965.10	8.59	+0.322	−1/+2	93	56	81
19	GMVLLHCSPALDFHKTPAV	2036.09	2035.04	6.91	+0.616	−1/+1	18	53	14
20	LPFLLGVGGLLGGSVGGGGGGGGAPL	2136.24	2135.17	5.52	+1.023	0/0	66	33	36
21	MVLDGKGGGGLLGGVLGGGKDAHLGG	2292.32	2291.21	6.50	+0.319	−2/+2	84.3	59.0	71
22	LLKDNGVGGLLGGGGAGGGGLVGGNLGGGAG	2478.39	2477.30	5.84	+0.439	−1/+1	86.4	54	66
23	KTSKLMVYLAGGGGGLLGGVGGGGGGAGGGGP GGL	2843.76	2842.48	9.70	+0.374	0/+2	76	47	67

The AASs of these peptides were identified previously by ^a^ Dolashki et al., 2018 [38]; ^b^ Dolashki et al., 2020 [21]; ^c^ Topalova et al., 2022 [39]; and ^d^ Velkova at al., 2018 [40].

**Table 2 molecules-29-02886-t002:** AASs of peptides, determined after the analysis of their MS/MS spectra. Proteins were identified after comparing AASs with a database of protein sequences by the Basic Local Alignment Search Tool (BLAST).

Band kDa	AAS of Peptide	Mass Exp. [M+H]^+^	Protein Name	UniProt ID	Identities
17.5	TAAFTEDTSVVTGR	1454.63	Mucus protein [*Cornu aspersum*]	QEG59314.1	86%, E = 8 × 10^−5^
	FTAAFTEDTSVAEGR	1601.74	Mucus protein [*C. aspersum*]	QEG59314.1	100%, E = 2 × 10^−9^
	GNVAFTAAFTEDTSVAEGR	1942.91	Mucus protein [C. aspersum]	QEG59314.1	100%, E = 4 × 10^−13^
	GALNGNVAFTAAFTEDTSVAEGR	2298.10	Mucus protein [*C. aspersum*]	QEG59314.1	100%, E = 2 × 10^−11^
23.6	GSCNWPMTILMLESLR	1850.90	NADH subunit 6 [*Albinaria caerulea*]	P48922	78%, E = 0.005
	QYLQITWPSPR	1388.7	H-type lectin domain-cont. protein *B. glabrata*	KAI8776186.1	100%, E = 0.041
26.7	VPSDDPGR	842.40	Chain A, *H. aspersa* agglutinin [*C. aspersum*]	4Q56_A	100%, E = 0.003
	DITFASPYCR	1172.54	Chain A, *H. aspersa* agglutinin *C. aspersum*	4Q56_A	90%, E = 1 × 10^−4^
	NGGLVHPGPR	1003.54	Uncharacterized protein [*Pomacea canaliculata*]	XP_025078819.1	89%, E = 4.5
31.35	DWTLYVNTPLAPAR	1616.84	Unnamed protein, partial [*C. unifasciata*] mucin 18b [*Plakobranchus ocellatus*]	CAG5130849.1 GFO09821.1	78%, E = 0.33 75%, E = 2.7
	FCPNAQRTR	1092.54	Glutathione S-transferase omega-1-like [*P. canaliculata*] Glutathione S-transferase omega-1 [*Elysia marginata*]	XP_025114871.1 GFR70608.1	89%, E = 0.31 89%, E = 0.31
	NPVGSVPVLELDGK	1423.78	Glutathione S-transferase omega-1 [*P. canaliculata*] Glutathione S-transferase omega-1-like [*B.glabrata*]	XP_025099831.1 KAI8762368.1	86%, E = 0.013 79%, E = 0.026
33.0	GPCFTPHTYTNWSWLR	1965.91	Unnamed protein [*Candidula unifasciata*] “Bacterial protein of unknown function” (HtrL_YibB)	CAG5121511.1	63%, E = 4 × 10^−4^
	DFLPPASLPDFAPSPPRVAER	2279.18	Unnamed protein product [*C. unifasciata*]	CAG5136685.1	53%, E = 0.34
39.1	AGYLQITWPSPR	1388.72	Mucus protein [*C. aspersum*] H-type lectin domain [*Biomphalaria glabrata*]	QEG59312.1 KAI8776186.1	100%, E = 1 × 10^−8^ 100% E = 0.055
	ASVTGDLSNK	991.51	Mucus protein [*C. aspersum*]	QEG59312.1	100%, E = 8 × 10^−4^
	ELLGNVYRAAFTEDTSVAEGR	2298.14	Mucus protein [*C. aspersum*]	QEG59314.1	77%, E = 3 × 10^−8^
	VGSNGAR	660.34	Mucus protein [*C. aspersum*]	QEG59312.1	100%, E = 0.004
	LPLYEDPKLDVSSLR	1744.83	Uncharacterized protein LOC101853718 [*A. californica*]	XP_012941625.1	83%, E = 0.27
48.7	FDANPFFSGR	1157.59	Hemocyanin, β_C_ chain unit D [*Helix pomatia*]	P12031.2	89%, E = 9 × 10^−6^
	HGSPIGVPYWDWTR	1670.93	Hemocyanin α N [*C. aspersum*]	AYO86684.1	100%, E = 2 × 10^−9^
	LISEATYFNSR	1300.71	Hemocyanin β [*C. aspersum*] β_C_ chain unit D [*H. pomatia*]	AYO86685.1, P12031.2	100%, E = 2 × 10^−4^ 82%, E = 6 × 10^−5^
	FDPNPFFSGR	1183.5	Hemocyanin, β_C_ chain unit D [*H. pomatia*]	P12031.2	100%, E = 3 × 10^−7^
	EVFEQVEHALLAR	1540.81	Hemocyanin β [*C. aspersum*] Hemocyanin β_C_ chain unit D [*H. pomatia*]	AYO86685.1 P12031.2	100%, E = 0.006 88%, E = 0.050
	QYLQITWSPPR	1388.73	Fibrinogen-like protein A isoform [*B. glabrata*]	XP_055864456.1	78%, E = 1.4
52.8	TDVTSALLGARCNEGGTNTHSPLR	2470.21	Unnamed protein, partial [*C. unifasciata*] Unnamed protein *C. unifasciata*	CAG5131631.1 CAG5131621.1	70%, E = 0.006 63%, E = 0.26
	LTQHFNVGSNGAR	1400.70	Mucus protein [*C. aspersum*] Unnamed protein [*C. unifasciata*] Unnamed protein partial [*C. unifasciata*]	QEG59312.1 CAG5131621.1 CAG5131631.1	92%, E = 0.001 77%, E = 0.094 75%, E= 0.38
	MPDYDCGCCNGSNGSYGSGGGGGGGR	2384.84	Unnamed protein, partial [*C.unifasciata*] Unnamed protein product [*C. unifasciata]*	CAG5126455.1 CAG5131834.1	80%, E = 0.23 63% E = 0.23
56.9	DGMSAAPQAENAFALK	1620.77	Achacin; flags: precursor [*L. fulica*] L-amino-acid oxidase (Escapin) [*A. californica*]	P35903.1 Q6IWZ0.1	71%, E = 3 × 10^−4^ 63%, E = 0.95
	SGFPTTSKDR	1095.54	L-amino-acid oxidase (Escapin) [*A. californica*]	Q6IWZ0.1	100%, E = 0.080
	VGGGPSGVELDEFEARVELK	2088.0	Achacin; flags: precursor [*Lissachatina fulica*]	P35903.1	41% E = 0.40
	AELKGDMQYTYADSEKVR	2104.00	Fibrillin-3, partial [*Biomphalaria pfeifferi*] Fibrillin-3, partial [*B. glabrata*]	KAK0048148.1 KAK6970092.1	48%, E = 0.22 48%, E = 0.22
70.8	AKVQVIGVPDDRLLLR	1792.08	Acyl-CoA synthetase family member 2, mitochondrial-like, partial [*Pomacea canaliculata*] Unnamed protein product [*C. unifasciata*]	XP_025086706.1 CAG5136482	91%, E = 0.005 91%, E = 0.005
	HGGGGGGFGGGGFGSR	1320.65	Uncharacterized protein LOC124113905 [*H. rufescens*] Fibroin heavy-chain isoform X1 [*A. californica*] Elastin-like [*Ostrea edulis*] Glycine, alanine asparagine-rich protein [*Haliotis asinina*]	XP_046330350.2 XP_012943828.1 XP_056017499.1 P86732.1	100%, E = 9 × 10^−4^ 81%, E = 0.018 87%, E = 0.038 65%, E = 0.004
84.7	YVLEDLSAADLELSR	1693.86	FMRFamide-activated amiloride-sensitive sodium channel [*C. aspersum*]	Q25011.1	100%, E = 0.14
	GSSVSLCVWDWAVLPR	1774.8	FMRFamide-activated amiloride-sensitive sodium channel [*C. aspersum*]	Q25011.1	71%, E = 1.9
97.0	MVTSLYPGEDWAMLPR	1865.89	Mucin-5AC-like, partial [*Haliotis rufescens*]	XP_048239711.1	56%, E = 0.15
	ETVSPGYSEGECTCASITQK	2089.91	Mucin-5B-like isoform X4 [*H. rubra*]	XP_046552239.1	62%, E = 6.7
	VFADFELHNIGASADVR	1860.92	Hemocyanin β [*C. aspersum*]	AYO86685.1	100%, E = 2 × 10^−10^
	CQVSLPYWDWAVPLR	1832.92	Hemocyanin, β_C_ chain unit G [*H. pomatia*]	P56823.1	100%, E = 5 × 10^−4^
	LSGYDVSKVNEILK	1564.86	Hemocyanin β [*C. aspersum*]	AYO86685.1	85%, E = 8 × 10^−5^
	HLWLLHCFEHDLNGYEYDNLR	2687.25	Hemocyanin β [*C. aspersum*]	AYO86685.1	87% E = 8 × 10^−6^
	ANNARLTDASHDNPFSSYTLR	2350.12	Hemocyanin β [C. *aspersum*]	AYO86685.1	94%, E = 1 × 10^−9^
143.3	VGRIESESYVVKKSK	1708.96	Zinc finger protein 502-like [*B. glabrata*] Zinc finger protein 184 [*B. glabrata*]	XP_055863608 KAI8795828.1	77%, E = 0.12 77%, E = 0.12
	VLTFYQSCDCVVDHCHR	2024.88	Zinc finger protein 664-like [*Patella vulgata*]	XP_050401404.1	75%, E = 1.5
	GEPGSTGPPGLK	1096.56	Collagen alpha-1(IV) chain-like, partial [*P. canaliculata*]	XP_025114158.1	100%, E = 2 × 10^−4^
	NGPSGVELDEFEARVELK	1988.99	Collagen alpha-1(XIII) chain-like isoform X7 [*H. rubra*]	XP_046573418.1	59%, E = 2.5
	FVDMDGMFSSAAGGRPEAR	2000.89	Collagen α-4(VI) chain-like X2 [*P. acuta*] Collagen α-1(XII) chain-like [*P. acuta*] Collagen α-1(XII) chain-like isoform X6 [*B. glabrata*]	XP_059154404.1 XP_059163536.1 XP_055860588.1	73%, E = 0.012 75%, E = 0.26 68%, E = 0.26
	RAYTDGMFSSAAGGRPEAR	1999.94	Collagen α-4(VI) chain-like isoform [*P. acuta*] Collagen α-4(VI) chain-like isoform [*P. acuta*]	XP_059154404.1 XP_059154396.1	79%, E = 0.031 79%, E = 0.031
	CNEGGTNTHSPLR	1385.62	Collagen α-6(VI) chain-like [*Physella acuta*]	XP_059164377.1	89%, E = 6.1
198.31	NPGYLVSKVNEILK	1573.89	Mucin-2-like [*P. acuta*] Serine/arginine repetitive matrix protein 2 [*B. glabrata*]	XP_059140993.1 KAI8735946.1	71%, E = 0.84 71%, E = 0.84
	DCRVRVGGPLFAPSPYLFER	2279.1	Mucin-17-like [*Haliotis rubra*]	XP_046554219.1	52%, E = 0.78
	FNFLPLSADALESLR	1692.90	E3 ubiquitin-protein ligase HERC2-like isoform X2 [*B. glabrata*]	XP_055894969.1	83%, E = 0.50
	ASSGSCDFSSSTAANR	1547.64	Mucin-5AC-like isoform X2 [*H. rufescens*] Mucin-5AC-like isoform X1 [*H. rufescens*]	XP_048252836.1 XP_048252834.1	85%, E = 0.45 85%, E = 0.45
	GCAGCPQAGSK	978.41	Fibrillin-1-like [*B. glabrata*]	XP_055888733.1	89%, E = 2.9

## Data Availability

Data are contained within the article and Appendix A.

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
