# Peer review of "Antibacterial Properties of Peptide and Protein Fractions from Cornu aspersum Mucus"

_molecules, 2024, doi:10.3390/molecules29122886_

Round 1

Reviewer 1 Report

Comments and Suggestions for Authors

The manuscript Antibacterial properties of peptide and protein fractions from the Cornu aspersum mucus is very well organized, has a very significant amount of work and a considerable variety of experimental and computational techniques. However, there is an issue that seems to me to be of substantial importance in the world of science, unlike the world of marketing: as scientists we must know how to report objectively and project, but without overvaluing unfounded results by extrapolation.

The work is very good and I do not want its contribution to be considered minor, but I am going to evaluated it as major concerns, because that is what they seem to me, in terms of the dissemination and conclusion of scientific work results.

MAJOR CONCERNS:

1) In the abstract of the manuscript, the authors provide wrong information regarding the data shown in the result section. Throughout the manuscript, they indicate that in the fraction with the highest MW, equivalent results have been found to the positive controls with vancomycin, but only at the highest concentrations. In the abstract, the authors equal the action of vancomycin with the more dilute C. aspersum mucus' fractions of higher MW .

2) In the same sentence of the abstract, regarding item 1, the authors indicate " without unwanted cytotoxic effects against non-target cells", this is an extrapolation of results that may be a projection, but is very far from being proven in this work and is written as a statement, which generates an expectation that is not found in the manuscript. The non-target cells are not defined, however the first sentence of the summary assumes that these products may not be toxic to humans. The fact that there is no lytic action over the yeasts does not allow the authors to extend this statement, without any type of clarification.

3) The authors attribute the antibacterial action of the mucus studied to its protein composition, however there are current studies that extend the same not only to the protein part, but also to carbohydrates. The writing of the manuscript must be reviewed so that one thing is not confused with the other. On the one hand, the lytic action is based on an extract and not on purified proteins. There are several paragraphs that generate confusion in this regard.

4) AMPs mostly show action on the membrane and it is true that they have a high tendency to present positive charges. However, the simplification made by the authors regarding the lipid composition of eukaryotic and prokaryotic membranes is far from adequate. It is true that the major lipids in the outer membrane of a large part of eukaryotic cells are PC and SM, nevertheless, the presence of species such as phosphatidylinositol, phosphatidylserine, phosphatidic acid and some negatively charged gangliosides invalidate this generalization. The lipid compositions of cells are very specific depending on their function and location, especially in complex organisms, and many AMPs also generate deleterious effects in many eukaryotic cells. Therefore, it is worth reviewing these types of oversimplifications.

MINOR CONCERNS:

1) There is text painted in yellow, in two places in the document.

2) My native language is not English, so I can only say that there are some words and forms of expression that are not common. I believe that the manuscript should be reviewed by a native English speaker to adequately resolve these expressions.

Comments on the Quality of English Language

My native language is not English, so I can only say that there are some words and forms of expression that are not common. I believe that the manuscript should be reviewed by a native English speaker to adequately resolve these expressions.

Author Response

ANSWERS TO THE REVIEWERS

We would like to thank the reviewers and the Editor for the time and effort taken to improve our manuscript. We have revised the manuscript according to all comments. We appreciate the opinions of the reviewers and agree with their comments. We have carefully considered the comments and tried our best to address every one of them. We hope the manuscript, after careful revisions, meets your high standards. The revised parts are marked in highlight in the revised manuscript.

Reviewer 1:

MAJOR CONCERNS:

1) In the abstract of the manuscript the authors provide wrong information regarding the data shown in the result section. Throughout, the manuscript, they indicate that in the fraction with the highest MW, equivalent results have been found to the positive controls with vancomycin, but only at the highest concentrations. In the abstract, the authors equal the action of vancomycin with the more dilute C. aspersum mucus fractions of higher MW.

Answer: Thank you for the comment. A technical mistake about the concentration was made in the abstract, which has been corrected in the revised version of the manuscript. All results demonstrated that the mucus fraction with MW> 20 kDa in concentrations between 32–128 µg/ml showed promising antimicrobial activity against the tested bacteria, comparable to Vancomycin.

 2) In the same sentence of the abstract, regarding item 1, the authors indicate " without unwanted cytotoxic effects against non-target cells", this is an extrapolation of results that may be a projection, but is very far from being proven in this work and is written as a statement, which generates an expectation that is not found in the manuscript. The non-target cells are not defined.

Answer:  Thank you for the comment. We fully agree with the proposed recommendations.

The cells of the 5 bacterial pathogens (Bacillus cereus 1085, Propionibacterium acnes 1897, Salmonella enterica 8691, Enterococcus faecalis 3915, and Enterococcus faecium 8754) are our target cells while eukaryotic cells of Saccharomyces cerevisiae cells are non-target cells. The abstract has been corrected in the revised version of the manuscript, as follows:

“The discovery and investigation of new natural compounds with antimicrobial activity is a potential strategy to reduce the spread of antimicrobial resistance. The presented study reveals for the first time the promising antibacterial potential of two fractions from Cornu aspersum mucus with MW<20 kDa and MW>20 kDa against five bacterial pathogens - Bacillus cereus 1085, Propionibacterium acnes 1897, Salmonella enterica 8691, Enterococcus faecalis 3915, and Enterococcus faecium 8754. Using de novo sequencing 16 novel peptides with potential antibacterial activity were identified in a fraction with MW<20 kDa. Some bioactive compounds in mucus fraction with MW > 20 kDa, were determined by proteomic analysis on 12% sodium dodecyl sulfate–polyacrylamide gel electrophoresis (SDS–PAGE) and bioinformatics. High homology with proteins and glycoproteins was found with potential antibacterial activity as mucus protein named Aspernin, hemocyanins, H-lectins, L-amino acid oxidase-like protein, and mucins (mucin-5AC, mucin-5B, mucin-2 and mucin-17). We hypothesize that the synergy between the bioactive components determined in the composition of the fraction > 20 kDa is responsible for the high antibacterial activity against the tested pathogens in concentrations between 32–128 µg/ml, which is comparable to Vancomycin, but without cytotoxic effects on model eukaryotic cells of Saccharomyces cerevisiae. Additionally, a positive effect by reducing the levels of intracellular oxidative damage and increasing the antioxidant capacity on S. cerevisiae cells was found for both mucus extract fractions of C. aspersum. These findings may serve as a basis for further studies to develop a new antibacterial agent preventing the development of antibiotic resistance.”

However, the first sentence of the summary assumes that these products may not be toxic to humans. The fact that there is no lytic action over the yeasts does not allow the authors to extend this statement without any type of clarification.

Answer:  S. cerevisiae cells are often used to study eukaryotic cell biology and as an ECHA model system for genetic toxicology assays. However, we agreed with the reviewer that more experiments are required to confirm the assumption made in the abstract that there is no human toxicity based only on experiments with yeast cells and the relevant texts in the conclusion were revised so that they correspond to the results obtained in the conducted experimental studies.

3) The authors attribute the antibacterial action of the mucus studied to its protein composition, however there are current studies that extend the same not only to the protein part but also to carbohydrates. The writing of the manuscript must be reviewed so that one thing is not confused with the other. On the one hand, the lytic action is based on an extract and not on purified proteins. There are several paragraphs that generate confusion in this regard.

Answer: Thank you for your comment. We completely agree with your comments.

Several studies in recent years have shown that mucus of land snails is a complex mixture of bioactive substances, primarily metabolites (free amino acids, glycolic acid, alantoin, glutathione and etc), antimicrobial peptides, proteins and glycoprotein (N-glycosylated proteins, lectins and mucins). The polysaccharides such as glycosaminoglycans (GAGs) also were identified in the mucus of land snails [Zhu et al., 2024; Deng et al., 2023; Kodchakorn et al., 2023; Trapella, et al., 2018; Gentili et al., 2020]. However, the results presented in these studies show that GAGs are not associated with antibacterial activity.

The study by Deng et al., 2023 commented that the mucus of snails Achatina fulica and Helix lucorum, consists of 30–50% protein and 10–16% sulfated glycosaminoglycan, and has a composition similar to that of the extracellular matrix. In this study it was found that the mucus from A. fulica contains 34.1% protein and 15.9% GAG, while the mucus from H. lucorum - 46.0% protein and 9.3% GAG. Deng et al., showed that GAGs effectively promoted chronic wound healing in the diabetic rat model by improving skin incision adhesion and accelerating granulation tissue regeneration, angiogenesis, and collagen deposition.

Furthermore, the study by Kodchakorn et al., 2023 shows that heparin-like sulfated glycosaminoglycan (named acharan sulfate) found in the mucus of A. fulica has potential application as an alternative to heparin. Moreover, antiviral properties have also been seen by the glycosaminoglycans produced by A. fulica. That a charan sulfate was able to inhibit the binding of SARS-CoV-2 spike protein to the ACE2 receptor, preventing the virus's ability to infect cells [Kodchakorn et al., 2023].

Studies on Helix aspersa muller mucus extracts (HelixComplex) also show the presence of GAGs (sulfurated and unsulfurated) [Trapella, et al., 2018; Gentili et al., 2020], which related to bioadhesive efficacy and the wound repair. HelixComplex has been found to promote cell migration and the wound healing processes both directly and indirectly [Trapella, et al., 2018].

These studies clearly show that the bioactive properties of glycosaminoglycan, found in the mucus of snails, are associated primarily promote wound healing processes, promising potential for the preparation of sulfated polysaccharides as an alternative to heparin and antiviral properties, but its antibacterial action there is no information yet.

Therefore, the additional paragraph is included in the manuscript:

“Several studies show that polysaccharides such as glycosaminoglycans (GAG) were also found in mucus of land snails [30,31,71-73]. Deng et al., (2023) showed that sulfated glycosaminoglycan in the mucus of A. fulica and H. lucorum effectively promoted chronic wound healing in the diabetic rat model by improving skin incision adhesion and accelerating granulation tissue regeneration, angiogenesis, and collagen deposition [31]. Recently Kodchakorn et al. (2023) reported that heparin-like sulfated glycosaminoglycan (named acharan sulfate) found in the mucus of A. fulica has promising potential for preparation of sulfated polysaccharides as an alternative to heparin and shows antiviral properties [71]. However, there is still no information about antibacterial action of the mucus glycosaminoglycans.

We hypothesize that the synergy between the bioactive components determined in its composition mainly proteins and glycoproteins such as mucus protein named Aspernin, N-glycosylated C. aspersum hemocyanin, lectins, L-amino acid oxidase-like protein, and mucins is responsible for the high antibacterial effect of the mucus fraction > 20 kDa comparable to Vancomycin against tested bacterial pathogens. ”  

  1. Zhu, K.; Zhang, Z.; Li, G.; Sun, J.; Gu, T.; Ain, N.U.; Zhang, X.; Li, D. Extraction, structure, pharmacological activities and applications of polysaccharides and proteins isolated from snail mucus. Int J Biol Macromol. 2024, 258(Pt 1), 128878. https://doi.org/10.1016/j.ijbiomac.2023.128878.
  2. Deng, T.; Gao, D.; Song, X.; Zhou, Z.; Zhou, L.; Tao, M.; Jiang, Z.; Yang, L.; Luo, L.; Zhou, A.; Hu, L.; Qin, H.; Wu, M. A natural biological adhesive from snail mucus for wound repair. Nat Commun 2023, 14, 396. https://doi.org/10.1038/s41467-023-35907-4
  3. Kodchakorn, K.; Chokepaichitkool, T.; Kongtawelert, P. Purification and characterisation of heparin-like sulfated polysaccharides with potent anti-SARS-CoV-2 activity from snail mucus of Achatina fulica. Carbohydr Res. 2023, 529, 108832. https://doi.org/10.1016/j.carres.2023.108832.
  4. Trapella, C.; Rizzo, R.; Gallo, S. et al. HelixComplex snail mucus exhibits pro-survival, proliferative and pro-migration effects on mammalian fibroblasts. Sci Rep 2018, 8, 17665. https://doi.org/10.1038/s41598-018-35816-3.
  5. Gentili, V.; Bortolotti, D.; Benedusi, M.; Alogna, A.; Fantinati, A.; Guiotto, A.; Turrin, G.; Cervellati, C.; Trapella, C.; Rizzo, R.; Valacchi, G. HelixComplex snail mucus as a potential technology against O3 induced skin damage. PLoS One. 2020, 15(2), e0229613. https://doi.org/10.1371/journal.pone.0229613.

The writing of the manuscript must be reviewed so that one thing is not confused with the other.

Answer: After reviewing the manuscript, we have made the necessary corrections.

On the one hand, the lytic action is based on an extract and not on purified proteins. There are several paragraphs that generate confusion in this regard.

Answer: In the present study, we have isolated two main fractions containing components with MW below 20 kDa and with MW above 20 kDa, which were tested for antibacterial activity against a panel of pathogenic bacteria. In order to explain the differences in the observed antibacterial activity of both fractions we have conducted studies for the characterization of bioactive compounds, contained in these fractions using mass spectrometry, electrophoretic analysis, proteomics and bioinformatics.  

In most studies concerning the biological activity (antimicrobial, antiviral, antitumor, regenerative, and anti-inflammatory properties) of the mucus of land snails, purified extracts are used [25,31,71-73] and/or fractions with different molecular weights [22, 23,26]. It has been shown that in most cases the biological activity of mucus is due to the complex action of various components. A recent study [72] has compared the effect of mucus extract from H. aspersa muller with the individual compounds allantoin and glycolic acid. The results have shown that both allantoin and glycolic acid alone did not have any regenerative properties. Therefore, the observed regenerative effect is a consequence of all the molecules present in the mucus and also of the specific ratio of which each component is present in the natural mucus. We found a promising concentration-dependent antibacterial effect of the two mucus fractions against the tested pathogenic bacterial strains, which may be due to the synergy between the components contained in them.

4) AMPs mostly show action on the membrane and it is true that they have a high tendency to present positive charges. However, the simplification made by the authors regarding the lipid composition of eukaryotic and prokaryotic membranes is far from adequate. It is true that the major lipids in the outer membrane of a large part of eukaryotic cells are PC and SM, nevertheless, the presence of species such as phosphatidylinositol, phosphatidylserine, phosphatidic acid and some negatively charged gangliosides invalidate this generalization. The lipid compositions of cells are very specific depending on their function and location, especially in complex organisms, and many AMPs also generate deleterious effects in many eukaryotic cells. Therefore, it is worth reviewing these types of oversimplifications.

Answer: We would like to thank the reviewer for the critical comments made. It is known that the amount of the negative charge in the prokaryotic cellular envelope is increased by the huge number of negative charges carried by the lipopolysaccharide (LPS) of Gram-negative bacteria and the lipoteichoic acid (LTA) of Gram-positive bacteria. However, phosphomannans and other membrane components, such as negatively charged phosphatidylinositol (PI), phosphatidylserine (PS), and diphosphatidylglycerol (DPG), can be responsible for the eukaryotic membranes' negative charges. Consequently, we agreed that the targeting specificity of cationic AMPs could only be partially explained by the preferential electrostatic interaction with the negatively charged bacterial membrane. With this respect, the introductory paragraph dedicated to the AMP's modes of action has been revised, providing more profound and specific information on the subject.

MINOR CONCERNS:

1) There is text painted in yellow, in two places in the document.

Answer: Apologies, this was overlooked on our behalf. It has been corrected.

2) My native language is not English, so I can only say that there are some words and forms of expression thate are not common. I believe that the manuscript should be reviewed by a native English speaker to adequately resolve these expressions.

Answer: Thank you for the recommendation, we have made a careful revision of the manuscript.

Reviewer 2 Report

Comments and Suggestions for Authors

The manuscript by Velkova and coauthors is a routine work on extraction and testing peptides from biological sources. It uses garden snails as peptide source. Peptides are identified by MS spectrometry, investigated through comparison with databases and tested on a set of pathogens.

The main drawback:

In the manuscript terms “mucus fractions” and “mucus extract” are used. Meanwhile in the experimental section no information on extraction technique is provided. Instead a patent is mentioned. Please provide valid reference to patent (either the patent number or application number) and the procedure as it is described in the patent. Also clearly indicate your relations with the patent and its owners in the Conflict of interests section. It is recommended that the term "mucus fractions" be replaced with "mucus extract fractions" if the the extraction had place.

Minors:

The acronym AR for antibiotic resistance is introduced in the introduction, but not used below. Instead, the AMR acronym is used. I advise replacing AMR with AR because AMR could be confused with AMP. The last is also used in the manuscript and stands for antimicrobial peptides.

Figure 2. Top right corner. What the number 1316.2 stands for? Is a mass, an intensity or a trace of data manipulation?

Figure 9. Errors bars are absent on the “a3” plot.

Comments on the Quality of English Language

English language is almost ok. Minor editing is required.

Author Response

ANSWERS TO THE REVIEWERS

We would like to thank the reviewers and the Editor for the time and effort taken to improve our manuscript. We have revised the manuscript according to all comments. We appreciate the opinion of the reviewers and agree with their comments. We have carefully considered the comments and tried our best to address every one of them. We hope the manuscript, after careful revisions, meets your high standards. The revised parts are marked in highlight in the revised manuscript.

Reviewer 2:

The main drawback:

In the manuscript terms “mucus fractions” and “mucus extract” are used. Meanwhile in the experimental section no information on extraction technique is provided. Instead a patent is mentioned. Please provide valid reference to patent (either the patent number or application number) and the procedure as it is described in the patent. Also, clearly indicate your relations with the patent and its owners in the Conflict of interests section. It is recommended that the term "mucus fractions" be replaced with "mucus extract fractions" if the extraction had place.

Answer:  Thank you for your comment and recommendations.

The mucus was collected from C. aspersa snails, grown in Bulgarian eco-farms by patented technology without disturbing their biological functions. A special device was used where the snails are placed with a small amount of distillate water and electrical stimulation with low voltage is applied. Thus obtained crude mucus extract was homogenized and centrifuged to remove coarse impurities. The mucus supernatant was subjected to several cycles of filtration, using filters of decreasing pore size for each subsequent filtration. The resulting native mucus extract was separated by ultrafiltration into different fractions based on molecular weight. The use of a non-invasive technique—ultrafiltration—ensured that we obtained fractions containing intact compounds.

In the experimental section 4. Materials and methods (4.1. Preparation of mucus extract) we have made the appropriate changes and placed a reference to the patent:

“The mucus was collected from C. aspersa snails, grown in Bulgarian eco-farms by patented technology without disturbing their biological functions [21,79].”

  1. Dolashka P., Atanasov D. Device for Collecting Extracts from Garden Snail. BG Utility model Application number 2656, 08.11.2013; Patent number 2097, 31.08.2015. (accessed on 19.05.2024). Available online: https://portal.bpo.bg/bpo_online/-/bpo/utility-model-detail

              Relations with the patent: Inventor and owner of the Useful model is prof. P. Dolashka, who is a corresponding author in this article, so there is no conflict of interest.

              In accordance with your recommendation, the term "mucus fractions" has been replaced with "mucus extract fractions", where appropriate in the text.

 Minors:

The acronym AR for antibiotic resistance is introduced in the introduction, but not used below. Instead, the AMR acronym is used. I advise replacing AMR with AR because AMR could be confused with AMP. The last is also used in the manuscript and stands for antimicrobial peptides.

Answer: Thank you for the suggestion. We used in the manuscript the acronym AR for antibiotic resistance. Necessary corrections have been made.

Figure 2. Top right corner. What the number 1316.2 stands for? Is a mass, an intensity or a trace of data manipulation?

Answer: The indicated value 1316.2 corresponds to the intensity of the most intense ion [M+H]+ et m/z 1438.8694, measured in [a.u.], in the MALDI-Tof-MS spectrum ( Figure 2 ) corresponds and corresponds to 100% Intensity.

Figure 9. Errors bars are absent on the “a3” plot.

Answer: The error bars in Figure 9 has been added.

Round 2

Reviewer 2 Report

Comments and Suggestions for Authors

The authors have not addressed the points I have raised.  The description of the methods used must be included in the "Experimental section" (not just in response to reviewer comments). The patent information (number or application number) must be included in the "conflict of interest" section.

Author Response

ANSWERS TO THE REVIEWERS

We would like to extend our sincerest apologies to the reviewers and the editor that the revised manuscript was not uploaded together with the reviewers' responses to the journal platform due to a system error that occurred.

Reviewer 2:

The authors have not addressed the points I have raised.  The description of the methods used must be included in the "Experimental section" (not just in response to reviewer comments). The patent information (number or application number) must be included in the "conflict of interest" section.

Answer: Dear Reviewer,

We would like to offer our sincerest apologies that due to an error in uploading the revised manuscript to the journal platform, you have not seen the corrections and additions made following your recommendations.

We have made a comprehensive revision of the manuscript in accordance with the recommendations you have made.

  • The abstract was completely corrected (lines 14-31; marked by highlight in yellow colour in the text).
  • In the section Introduction, we have made corrections and additions (paragraph between lines 49-72; marked by highlight in yellow colour in the text).
  • In the Discussion section we also have made corrections and additions (marked by highlight in yellow colour).
  • According to your recommendations, the description of the methods used is included in the experimental section: 1. Preparation of mucus extract:

“4.1. Preparation of mucus extract.

The mucus was collected from C. aspersa snails, grown in Bulgarian eco-farms by patented technology without disturbing their biological functions [21,79]. The method for collecting native extract from garden snails based on electrical stimulation with low voltage can be used repeatedly to extract mucus. Thus, obtained crude mucus extract was homogenized and centrifugated to remove coarse impurities. The supernatant was subjected to several cycles of filtration, using filters of decreasing pore size for each subsequent filtration. The protein concentration in the native mucus extract was determined by Bradford assay [80]. The obtained crude mucus extract was divided into two fractions by ultrafiltration, using membranes with pore sizes of 20 kDa (polyethersulfone, Microdyn Nadir™ from STERLITECH Corporation, Goleta, CA, USA, respectively): a peptide fraction with MW below 20 kDa and a fraction containing compounds with MW above 20 kDa.

The peptide fraction with MW < 20 kDa was additionally separated into three fractions using Amicon® Ultra-15 centrifugal tube filters with 3 and 10 kDa membranes. Finally, the following samples were obtained: Sample 1— a fraction with compounds of MW <3 kDa; Sample 2— a fraction with compounds of MW 3–10 kDa; Sample 3 fraction with compounds of Mw 10–200 kDa.

The use of a non-invasive technique—ultrafiltration—ensured that we obtained fractions containing intact compounds.”

  • In the section Conflicts of Interest, we have made an addition in relation to your comment:

“The authors declare no conflicts of interest. The funders had no role in the design of the study; in the collection, analysis, or interpretation of data; in the writing of the manuscript; or in the decision to publish the results.

In this manuscript is used BG Utility model: Dolashka, P.; Atanasov, D. Device for Collecting Ex-tracts from Garden Snail. BG Utility model Application number 2656, 08.11.2013; Patent number 2097, 31.08.2015. (accessed on 19.05.2024). Available online: https://portal.bpo.bg/bpo_online/-/bpo/utility-model-detail. Prof. P. Dolashka is both the inventor and the patent owner, but also she is a corresponding author in this article, so there is no conflict of interest.”

  • Following your recommendation, the term "mucus fractions" has been replaced with "mucus extract fractions" or "fractions extracted from the mucus of aspersum snail", where appropriate. We made more than 10 changes in the text, marked by highlights in grey colour.

Round 3

Reviewer 2 Report

Comments and Suggestions for Authors

Authors had addressed my comments.

Comments on the Quality of English Language

The English should be improoved. For example, the following sentence in the Methods section is incorrect.

"The method for col608 lecting native extract from garden snail on based electrical stimulation with low voltage was 609 used can be used repeatedly to extract mucus"